# Jeffreys Divergence and Generalized Fisher Information Measures on Fokker–Planck Space–Time Random Field

**DOI:** 10.3390/e25101445

**Published:** 2023-10-13

**Authors:** Jiaxing Zhang

**Affiliations:** School of Mathematics, Tianjin University, Tianjin 300350, China; 2017233010@tju.edu.cn; Tel.: +86-1512-750-2081

**Keywords:** space–time random field, Fokker–Planck equations, differential entropy, Jeffreys divergence, fisher information, De Bruijn identities

## Abstract

In this paper, we present the derivation of Jeffreys divergence, generalized Fisher divergence, and the corresponding De Bruijn identities for space–time random field. First, we establish the connection between Jeffreys divergence and generalized Fisher information of a single space–time random field with respect to time and space variables. Furthermore, we obtain the Jeffreys divergence between two space–time random fields obtained by different parameters under the same Fokker–Planck equations. Then, the identities between the partial derivatives of the Jeffreys divergence with respect to space–time variables and the generalized Fisher divergence are found, also known as the De Bruijn identities. Later, at the end of the paper, we present three examples of the Fokker–Planck equations on space–time random fields, identify their density functions, and derive the Jeffreys divergence, generalized Fisher information, generalized Fisher divergence, and their corresponding De Bruijn identities.

## 1. Introduction

Information entropy and Fisher information are quantities to measure random information, and entropy divergence is derived from information entropy to measure the difference between two probability distributions. Formally, we can construct straightforward definitions of entropy divergence and Fisher information for the case of a space–time random field found on classical definitions. The density function, in their definitions, can be obtained in many different ways. In this paper, the density function of a space–time random field is obtained by Fokker–Planck equations. The traditional Fokker–Planck equation is a partial differential equation that describes the probability density function of a random process [1]. It describes the density function’s time-varying change rule. However, the Fokker–Planck equations for random fields, especially for space–time random fields, do not yet possess a distinct form. The classical equation needs to be generalized because the variable varies from time to space–time.

In this paper, we mainly obtain the relation between Jeffreys divergence and generalized Fisher information measure for space–time random field generated by Fokker–Planck equations. Jeffreys divergence is a symmetric entropy divergence, which is generalized from Kullback–Leibler divergence (KL divergence). Jeffreys divergence is a measure in information theory and statistics that evaluates the variation between anticipated and real probability distributions. However, if there is no overlap between the two distributions, the outcome will be infinite, which is a limitation of this approach. To prevent infinite results, we examine how Jeffreys divergence relates to generalized Fisher information for a space–time random field with slight variations in space–time parameters.

Moreover, the classical De Bruijn identity describes the relationship between differential entropy and the Fisher information of the Gaussian channel [2], and it can be generalized to other cases [3,4,5,6,7]. With gratitude to their works and following their ideas, we obtain De Bruijn identities on Jeffreys divergence and generalized Fisher information of space–time random fields, whose density functions satisfy Fokker–Planck equations.

### 1.1. Space–Time Random Field

The random field was first studied by Kolmogorov [8,9,10], and it was gradually improved by Yaglom [11,12,13] in the middle of the last century. A random field with $n\in N_{+}$ variables can be expressed as
(1)$$X(t_{1},t_{2},\cdots ,t_{n})$$
where $\left(t_{1},t_{2},\cdots ,t_{n}\right)\in R^{n}$. We call (Equation 1) a generalized random field or a multiparameter stochastic process. In some practical applications, we often use the concept of space–time random field. The space–time random field on a *d*-dimensional space is expressed as
(2)X(t,x)
where $\left(t,x\right)\in R_{+}\times R^{d}$ are the space–time variables. It has many applications in statistics, finance, signal processing, stochastic partial differential equations, and other fields [14,15,16,17,18,19,20,21,22,23,24,25,26,27].

### 1.2. Kramers–Moyal Expansion and Fokker–Planck Equation

In the literature of stochastic processes, Kramers–Moyal expansion refers to a Taylor series of the master equation, named after Kramers and Moyal [28,29]. The Kramers–Moyal expansion is an infinite order partial differential equation
(3)$$\frac{\partial }{\partial t}p\left(u,t\right)=\sum_{n=1}^{\infty }\frac{{\left(-1\right)}^{n}}{n!}\frac{\partial^{n}}{\partial u^{n}}\left[K_{n}\left(u,t\right)p\left(u,t\right)\right]$$
where p(u,t) is the density function and
(4)$$K_{n}\left(u,t\right)=\int_{R}{\left(u^{\prime }-u\right)}^{n}W\left(u^{\prime }|u,t\right)du^{\prime }$$
is the *n*-order conditional moment. Here, $W(u^{\prime }|u,t)$ is the transition probability rate. The Fokker–Planck equation is obtained by keeping only the first two terms of the Kramers–Moyal expansion. In statistical mechanics, the Fokker–Planck equation is usually used to describe the time evolution of the probability density function of the velocity of a particle under the influence of drag forces and random forces, as in the famous Brownian motion, and this equation is commonly employed for determining the density function of an Itô stochastic differential equation [1].

### 1.3. Differential Entropy and De Bruijn Identity

The entropy of a continuous distribution was proposed by Shannon in 1948, known as differential entropy [30]:(5)$$h\left(X\right)=-\int_{R^{d}}p\left(x\right)\log p\left(x\right)dx$$
where h(·) represents the differential entropy and p(·) is the probability density function of *X*. However, differential entropy is not easy to calculate and seldom exists. There are related studies on the entropy of stochastic processes and continuous systems [31,32,33,34]. If we consider a classical one-dimensional Gaussian channel model
(6)$$Y_{t}=X+\sqrt{t}G$$
where *X* is the input signal, *G* is standard Gaussian noise, t≥0 is the strength, and $Y_{t}$ is the output, we can obtain that the density of $Y_{t}$ satisfies the following Fokker–Planck equation:(7)$$\frac{\partial }{\partial t}p\left(y,t\right)=\frac{1}{2}\frac{\partial^{2}}{\partial y^{2}}p\left(y,t\right)$$

Furthermore, the differential entropy of $Y_{t}$ can be calculated, and then its derivative with respect to *t* can be obtained as
(8)$$\frac{dh_{Y_{t}}\left(t\right)}{dt}=\frac{1}{2}FI_{Y_{t}}\left(t\right)$$
where
(9)$$FI_{Y_{t}}\left(t\right)=\int_{R}{\left[\frac{\partial \log p(y,t)}{\partial y}\right]}^{2}p\left(y,t\right)dy$$
is the Fisher information of $Y_{t}$. The Equation (Equation 8) here is the De Burijn identity. The de Bruijn identity connects the differential entropy h(·) and the Fisher information FI(·), which shows that they are different aspects of the concept of “information”.

### 1.4. Entropy Divergence

In information theory and statistics, an entropy divergence is a statistical distance generated from information entropy to measure the difference between two probability distributions. There are various divergences generated by information entropy, such as Kullback–Leibler divergence [35], Jeffreys divergence [36], Jensen-Shannon divergence [37], and Rényi divergence [38]. These measures are applied in a variety of fields such as finance, economics, biology, signal processing, pattern recognition, and machine learning [39,40,41,42,43,44,45,46,47,48,49]. In this paper, we mainly focus on the Jeffreys divergence of two distributions, formed as
(10)$$JD\left(P,Q\right)=\int_{R}\left[p(u)-q(u)\right]\log\frac{p(u)}{q(u)}d\mu \left(u\right)$$
where μ is a measure of *u*.

## 2. Notations, Definitions, and Propositions

### 2.1. Notations and Assumptions

In this paper, we use the subsequent notations and definitions
•Given a probability space (Ω,F,P), two real valued space–time random fields are denoted as X(ω;t,x), Y(ω;s,y) or X(t,x), Y(s,y), where ω∈Ω and $\left(t,x\right)\;\left(s,y\right)\in R_{+}\times R^{d}$, $d\in N_{+}$ are space–time variables.•The probability density functions of *P* and *Q* are denoted as *p* and *q*. With ∀u∈R, p(u;t,x) is the density value at (t,x) of *X* and q(u;s,y) is the density value at (s,y) of *Y*.•Unless there are specific restrictions on the ranges of variables, suppose that our density functions p(u;t,x) and q(u,s,y) belongs to $C^{2,1,1}\left(R\times R_{+}\times R^{d},R\right)$. This means that p(u;t,x) and q(u;s,y) are partial differentiable twice with respect to *u* and once with respect to (t,x) or (s,y), respectively.•Vectors that differ only from the *k*-th coordinate of $x=(x_{1},x_{2},\cdots ,x_{k},\cdots ,x_{d})$ are denoted ${\tilde{x}}^{\left(k\right)}=\left(x_{1},x_{2},\cdots ,x_{k}^{\prime },\cdots ,x_{d}\right)$, where the *k*-th coordinates are $x_{k}$ and $x_{k}^{\prime }$, k=1,2,⋯,d.

### 2.2. Definitions

To obtain the generalized De Bruijn identities between Jeffreys divergence and Fisher divergence, we need to introduce some new definitions and propositions.

The primary and most important measure of information is the Kullback–Leibler divergence for random fields. Definition 1 is easily obtained as follows.

**Definition** **1.**
*The Kullback–Leibler divergence between two space–time random fields X(t,x) and Y(s,y), $\left(t,x\right),\;\left(s,y\right)\in R_{+}\times R^{d}$, with density functions p(u;t,x) and q(u;s,y), is defined as*

(11)
$$KL\left(P(t,x)∥Q(s,y)\right)=\int_{R}p\left(u;t,x\right)\log\frac{p(u;t,x)}{q(u;s,y)}du$$



Similar to the classical Kullback–Leibler divergence, Kullback–Leibler divergence on random fields is not symmetrical, i.e.,
(12)$$KL\left(P(t,x)∥Q(s,y)\right)\neq KL\left(Q(s,y)∥P(t,x)\right)$$

Following the classical definition of Jeffreys divergence on two random variables, we mainly consider Jeffreys divergence for random fields in this paper.

**Definition** **2.**
*The Jeffreys divergence between space–time random fields X(t,x) and Y(s,y), $\left(t,x\right),\;\left(s,y\right)\in R_{+}\times R^{d}$, with density function p(u;t,x) and q(u;s,y) is defined as*

(13)
$$JD\left(P(t,x),Q(s,y)\right)=KL\left(P(t,x)∥Q(s,y)\right)+KL\left(Q(s,y)∥P(t,x)\right)$$



Here, we replace ∥ with, in the distortion measure to emphasize the symmetric property.

Another significant measure of information is Fisher information. In this paper, we consider the generalized Fisher information of the space–time random field.

**Definition** **3.**
*The Generalized Fisher information of the space–time random field X(t,x), $\left(t,x\right)\in R_{+}\times R^{d}$, with density function p(u;t,x) defined by nonnegative function f(·), is formed as*

(14)
$$FI_{f}\left(P(t,x)\right)=\int_{R}f\left(u\right){\left[\partial_{u}\log p\left(u;t,x\right)\right]}^{2}p\left(u;t,x\right)du$$

*In this case, where f is equal to 1, $FI_{1}\left(P\left(t,x\right)\right)$ represents the typical Fisher information. In addition to Equation (Equation 14), there are similar forms of generalized Fisher information*

(15)
$$FI_{f}^{\left(t\right)}\left(P(t,x)\right)=\int_{R}f\left(u\right){\left[\partial_{t}\log p\left(u;t,x\right)\right]}^{2}p\left(u;t,x\right)du$$

*and*

(16)
$$FI_{f}^{\left(x_{k}\right)}\left(P(t,x)\right)=\int_{R}f\left(u\right){\left[\partial_{x_{k}}\log p\left(u;t,x\right)\right]}^{2}p\left(u;t,x\right)du$$

*for k=1,2,⋯,d.*


Obviously, (Equation 15) and (Equation 16) are generalized Fisher information on space–time variables. Regarding the generalized Fisher information (Equation 14), we can come to a following simple proposition.

**Proposition** **1.**
*For the arbitrary positive continuous function f(·), suppose the generalized Fisher information of continuous random variable X*

(17)
$$FI_{f}\left(X\right):=\int_{R}f\left(u\right){\left[\frac{d\log p_{X}\left(u\right)}{du}\right]}^{2}p_{X}\left(u\right)du$$

*is well defined, where $p_{X}\left(u\right)$ represents the probability density. Then, we have the generalized Fisher information inequality*

(18)
$$\frac{1}{FI_{f}\left(X+Y\right)}\geq \frac{1}{FI_{f}\left(X\right)}+\frac{1}{FI_{f}\left(Y\right)}$$

*when f≡1, $FI_{1}\left(X\right)$ represents the Fisher information in the standard case.*


**Proof.** Denote Z=X+Y, $p_{X}$, $p_{Y}$, and $p_{Z}$ represent densities, i.e.,
(19)$$p_{Z}\left(z\right)=\int_{R}p_{X}\left(x\right)p_{Y}\left(z-x\right)dx$$
and derivative function
(20)$$p_{Z}^{\prime }\left(z\right)=\int_{R}p_{X}^{\prime }\left(x\right)p_{Y}\left(z-x\right)dx$$If $p_{X}$, $p_{Y}$, and $p_{Z}$ never vanish,
(21)$$\begin{array}{cc} \; & \frac{p_{Z}^{\prime }\left(z\right)}{p_{Z}\left(z\right)}\\ = & \int_{R}\frac{p_{X}^{\prime }\left(x\right)p_{Y}\left(z-x\right)}{p_{Z}\left(z\right)}dx\\ = & \int_{R}\frac{p_{X}\left(x\right)p_{Y}\left(z-x\right)}{p_{Z}\left(z\right)}\frac{p_{X}^{\prime }\left(x\right)}{p_{X}\left(x\right)}dx\\ = & E\left[\frac{p_{X}^{\prime }\left(x\right)}{p_{X}\left(x\right)}|Z=z\right] \end{array}$$
is the conditional expectation of $\frac{p_{X}^{\prime }\left(x\right)}{p_{X}\left(x\right)}$ for given *z*. Similarly, we can obtain
(22)$$\frac{p_{Z}^{\prime }\left(z\right)}{p_{Z}\left(z\right)}=E\left[\frac{p_{Y}^{\prime }\left(y\right)}{p_{Y}\left(y\right)}|Z=z\right]$$
and ∀μ, λ∈R, we also find that
(23)$$E\left[\mu \frac{p_{X}^{\prime }\left(x\right)}{p_{X}\left(x\right)}+\lambda \frac{p_{Y}^{\prime }\left(y\right)}{p_{Y}\left(y\right)}|Z=z\right]=\left(\mu +\lambda \right)\frac{p_{Z}^{\prime }\left(z\right)}{p_{Z}\left(z\right)}$$Then, we have
(24)$$\begin{array}{cc} \; & {\left[\left(\mu +\lambda \right)\frac{p_{Z}^{\prime }\left(z\right)}{p_{Z}\left(z\right)}\right]}^{2}\\ = & {\left\{E\left[\mu \frac{p_{X}^{\prime }\left(x\right)}{p_{X}\left(x\right)}+\lambda \frac{p_{Y}^{\prime }\left(y\right)}{p_{Y}\left(y\right)}|Z=z\right]\right\}}^{2}\\ \leq & E\left\{{\left[\mu \frac{p_{X}^{\prime }\left(x\right)}{p_{X}\left(x\right)}+y\frac{p_{Y}^{\prime }\left(y\right)}{p_{Y}\left(y\right)}\right]}^{2}|Z=z\right\} \end{array}$$
with equality only if
(25)$$\mu \frac{p_{X}^{\prime }\left(x\right)}{p_{X}\left(x\right)}+\lambda \frac{p_{Y}^{\prime }\left(y\right)}{p_{Y}\left(y\right)}=\left(\mu +\lambda \right)\frac{p_{Z}^{\prime }\left(z\right)}{p_{Z}\left(z\right)}$$
with probability 1 whenever z=x+y and we have
(26)$$f\left(z\right){\left[\left(\mu +\lambda \right)\frac{p_{Z}^{\prime }\left(z\right)}{p_{Z}\left(z\right)}\right]}^{2}\leq f\left(z\right)E\left[{\left(\mu \frac{p_{X}^{\prime }\left(x\right)}{p_{X}\left(x\right)}+y\frac{p_{Y}^{\prime }\left(y\right)}{p_{Y}\left(y\right)}\right)}^{2}|Z=z\right]$$Averaging both sides over the distribution of *z*
(27)$$\begin{array}{cc} \; & E\left\{f\left(z\right){\left[\left(\mu +\lambda \right)\frac{p_{Z}^{\prime }\left(z\right)}{p_{Z}\left(z\right)}\right]}^{2}\right\}\\ \leq & E\left\{f\left(z\right)E\left[{\left(\mu \frac{p_{X}^{\prime }\left(x\right)}{p_{X}\left(x\right)}+y\frac{p_{Y}^{\prime }\left(y\right)}{p_{Y}\left(y\right)}\right)}^{2}|Z=z\right]\right\}\\ = & \mu^{2}E\left\{f\left(z\right)E\left[{\left(\frac{p_{X}^{\prime }\left(x\right)}{p_{X}\left(x\right)}\right)}^{2}|Z=z\right]\right\}+\lambda^{2}E\left\{E\left[f\left(z\right){\left(\frac{p_{Y}^{\prime }\left(y\right)}{p_{Y}\left(y\right)}\right)}^{2}|Z=z\right]\right\} \end{array}$$
i.e.,
(28)$${\left(\mu +\lambda \right)}^{2}FI_{f}\left(X+Y\right)\leq \mu^{2}FI_{f}\left(X\right)+\lambda^{2}FI_{f}\left(Y\right)$$Let $\mu =\frac{1}{FI_{f}\left(X\right)}$ and $\lambda =\frac{1}{FI_{f}\left(Y\right)}$, we obtain
(29)$$\frac{1}{FI_{f}\left(X+Y\right)}\geq \frac{1}{FI_{f}\left(X\right)}+\frac{1}{FI_{f}\left(Y\right)}$$□

According to Definition 3, we can obtain relevant definitions on the generalized Fisher information measure.

**Definition** **4.**
*The generalized Cross–Fisher information for space–time random fields X(t,x) and Y(s,y), $\left(t,x\right),\;\left(s,y\right)\in R_{+}\times R^{d}$, with density functions p(u;t,x) and q(u;s,y), defined by the nonnegative function f(·), is defined as*

(30)
$$CFI_{f}\left(P\left(t,x\right),Q\left(s,y\right)\right)=\int_{R}f\left(u\right){\left[\partial_{u}\log q\left(u;s,y\right)\right]}^{2}p\left(u;t,x\right)du$$



Similar to the concept of cross-entropy, it is easy to verify that (Equation 30) is symmetrical about *P* and *Q*.

**Definition** **5.**
*The generalized Fisher divergence for space–time random fields X(t,x) and Y(s,y), for $\left(t,x\right),\;\left(s,y\right)\in R_{+}\times R^{d}$, with density functions p(u;t,x) and q(u;s,y), defined by nonnegative function f(·), is defined as*

(31)
$$FD_{f}\left(P(t,x)∥Q(s,y)\right)=\int_{R}f\left(u\right){\left[\partial_{u}\log p\left(u;t,x\right)-\partial_{u}\log q\left(u;s,y\right)\right]}^{2}p\left(u;t,x\right)du$$

*In particular, when f≡1, $FD_{1}\left(P(t,x)∥Q(s,y)\right)$ represents the typical Fisher divergence.*


Obviously, the generalized Fisher divergence between two random fields is not a symmetrical measure of information. We need to create a new formula to expand on (Equation 31) in order to achieve symmetry.

**Definition** **6.**
*The generalized Fisher divergence for space–time random fields X(t,x) and Y(s,y), $\left(t,x\right),\;\left(s,y\right)\in R_{+}\times R^{d}$, with density functions p(u;t,x) and q(u;s,y), defined by nonnegative functions f(·) and g(·), is defined as*

(32)
$$\begin{array}{cc} FD_{\left(f,g\right)}\left(P(t,x)∥Q(s,y)\right)= & \int_{R}\left[f\left(u;t,x\right)\partial_{u}\log p\left(u;t,x\right)-g\left(u;s,y\right)\partial_{u}\log q\left(u;s,y\right)\right]\\ \; & \times \left[\partial_{u}\log p\left(u;t,x\right)-\partial_{u}\log q\left(u;s,y\right)\right]\left[p(u;t,x)+q(u,s,y)\right]du \end{array}$$

*In particular, if f equals g, the generalized Fisher divergence for random fields using a single function is denoted as $FD_{\left(f,f\right)}\left(P(t,x)∥Q(s,y)\right)$.*


In general, $FD_{\left(f,g\right)}\left(P(t,x)∥Q(s,y)\right)$ is asymmetric with respect to *P* and *Q*, i.e.,
(33)$$FD_{\left(f,g\right)}\left(P(t,x)∥Q(s,y)\right)\neq FD_{\left(f,g\right)}\left(Q(s,y)∥P(t,x)\right)$$

If we suppose that *f* and *g* are functions only related to *P* and *Q*, i.e.,
(34)$$\left\{\begin{array}{cc} \; & f\left(u;t,x\right)=T\left[p(t,x)\right]\left(u\right)\\ \; & g\left(u;s,y\right)=T\left[q(s,y)\right]\left(u\right) \end{array}\right.$$
where T is an operator; the generalized Fisher divergence $FD_{\left(f,g\right)}\left(P(t,x)∥Q(s,y)\right)$ can be rewritten as
(35)$$\begin{array}{cc} \; & FD_{\left(f,g\right)}\left(P(t,x)∥Q(s,y)\right)\\ = & \int_{R}\left\{T\left[p(t,x)\right]\left(u\right)\partial_{u}\log p\left(u;t,x\right)-T\left[q(s,y)\right]\left(u\right)\partial_{u}\log q\left(u;s,y\right)\right\}\\ \; & \times \left[\partial_{u}\log p\left(u;t,x\right)-\partial_{u}\log q\left(u;s,y\right)\right]\left[p(u;t,x)+q(u,s,y)\right]du \end{array}$$
and we can easily obtain
(36)$$FD_{\left(f,g\right)}\left(P(t,x)∥Q(s,y)\right)=FD_{\left(g,f\right)}\left(Q(t,x)∥P(s,y)\right)$$

In this case, we call (Equation 35) symmetric Fisher divergence for random fields generated by operator T and denote it as
(37)$$\begin{array}{cc} \; & sFD_{T}\left(P(t,x),Q(s,y)\right)\\ = & \int_{R}\left\{T\left[p(t,x)\right]\left(u\right)\partial_{u}\log p\left(u;t,x\right)-T\left[q(s,y)\right]\left(u\right)\partial_{u}\log q\left(u;s,y\right)\right\}\\ \; & \times \left[\partial_{u}\log p\left(u;t,x\right)-\partial_{u}\log q\left(u;s,y\right)\right]\left[p(u;t,x)+q(u,s,y)\right]du \end{array}$$Notice that
(38)$$\begin{array}{cc} \; & Aa-Bb\\ = & \frac{1}{2}\times 2\left(Aa-Bb\right)\\ = & \frac{1}{2}\left[\left(Aa-Ab+Ab-Bb\right)+\left(Aa-Ba+Ba-Bb\right)\right]\\ = & \frac{1}{2}\left[\left(A+B\right)\left(a-b\right)+\left(A-B\right)\left(a+b\right)\right] \end{array}$$
for *A*, *B*, *a*, b∈R; then, we can rewrite (Equation 37) as
(39)$$\begin{array}{cc} \; & sFD_{T}\left(P(t,x),Q(s,y)\right)\\ = & \frac{1}{2}\int_{R}\left\{T\left[p(t,x)\right]\left(u\right)+T\left[q(s,y)\right]\left(u\right)\right\}{\left[\partial_{u}\log p\left(u;t,x\right)-\partial_{u}\log q\left(u;s,y\right)\right]}^{2}\\ \; & \;\times \left[p(u;t,x)+q(u,s,y)\right]du\\ \; & +\frac{1}{2}\int_{R}\left\{T\left[p(t,x)\right]\left(u\right)-T\left[q(s,y)\right]\left(u\right)\right\}\left\{{\left[\partial_{u}\log p\left(u;t,x\right)\right]}^{2}-{\left[\partial_{u}\log q\left(u;s,y\right)\right]}^{2}\right\}\\ \; & \;\times \left[p(u;t,x)+q(u,s,y)\right]du\\ = & \frac{1}{2}\left[FD_{T\left[p(t,x)\right]+T\left[q(s,y)\right]}\left(P(t,x)∥Q(s,y)\right)+FD_{T\left[p(t,x)\right]+T\left[q(s,y)\right]}\left(Q(s,y)∥P(t,x)\right)\right]\\ \; & +\frac{1}{2}\left[FI_{T\left[p(t,x)\right]-T\left[q(s,y)\right]}\left(P(t,x)\right)+FI_{T\left[p(t,x)\right]-T\left[q(s,y)\right]}\left(Q(s,y)\right)\right]\\ \; & +\frac{1}{2}\left[CFI_{T\left[p(t,x)\right]-T\left[q(s,y)\right]}\left(Q(s,y),P(t,x)\right)-CFI_{T\left[p(t,x)\right]-T\left[q(s,y)\right]}\left(P(t,x),Q(s,y)\right)\right] \end{array}$$

**Lemma** **1***(Kramers–Moyal expansion* [28,29]*)*. *Suppose that the random process X(t) has any order moment; then, the probability density function p(u,t) satisfies the Kramers–Moyal expansion*
(40)$$\frac{\partial }{\partial t}p\left(u,t\right)=\sum_{n=1}^{\infty }\frac{{\left(-1\right)}^{n}}{n!}\frac{\partial^{n}}{\partial u^{n}}\left[K_{n}\left(u,t\right)p\left(u,t\right)\right]$$
*where*
(41)$$K_{n}\left(u,t\right)=\int_{R}{\left(u^{\prime }-u\right)}^{n}W\left(u^{\prime }|u,t\right)du^{\prime }$$*is the n-order conditional moment and $W(u^{\prime }|u,t)$ is the transition probability rate.*

**Lemma** **2***(Pawula theorem *[50,51]*).**If the limit on conditional moment of random process X(t)*(42)$$\lim_{\Delta t\to 0}\frac{1}{\Delta t}E\left\{{\left[X(t+\Delta t)-X(t)\right]}^{n}|X\left(t\right)=x\right\}$$*exists for all $n\in N_{+}$, and the limit value equals 0 for some even number, then the limit values are 0 for all n≥3.*

The Pawula theorem states that there are only three possible cases in the Kramers–Moyal expansion:(1)The Kramers–Moyal expansion is truncated at n=1, meaning that the process is deterministic;(2)The Kramers–Moyal expansion stops at n=2, with the resulting equation being the Fokker–Planck equation, and describes diffusion processes;(3)The Kramers–Moyal expansion contains all the terms up to n=∞.

In this paper, we only focus on the case of the Fokker–Planck equation.

## 3. Main Results and Proofs

In this section, we establish the Fokker–Planck equations for continuous space–time random field. Additionally, we present the relationship theorem between Jeffreys divergence and Fisher information, as well as the De Bruijn identities connection between Jeffreys divergence and Fisher divergence.

**Theorem** **1.**
*The probability density function p(u;t,x) of the continuous space–time random field X(t,x), u∈R, $\left(t,x\right)\in R_{+}\times R^{d}$ satisfies the following Fokker–Planck equations:*

(43)
$$\left\{\begin{array}{cc} \frac{\partial }{\partial t}p\left(u;t,x\right) & =\frac{1}{2}\frac{\partial^{2}}{\partial u^{2}}\left[b_{0}\left(u;t,x\right)p\left(u,t,x\right)\right]-\frac{\partial }{\partial u}\left[a_{0}\left(u;t,x\right)p\left(u;t,x\right)\right]\\ \frac{\partial }{\partial x_{k}}p\left(u;t,x\right) & =\frac{1}{2}\frac{\partial^{2}}{\partial u^{2}}\left[b_{k}\left(u;t,x\right)p\left(u,t,x\right)\right]-\frac{\partial }{\partial u}\left[a_{k}\left(u;t,x\right)p\left(u;t,x\right)\right]\\ k & =1,2,\cdots ,d \end{array}\right.$$

*where*

(44)
$$\left\{\begin{array}{cc} a_{0}\left(u;t,x\right) & =\lim_{\Delta t\to 0}\frac{1}{\Delta t}M_{1}\left(u;t,\Delta t,x\right)\\ b_{0}\left(u;t,x\right) & =\lim_{\Delta t\to 0}\frac{1}{\Delta t}M_{2}\left(u;t,\Delta t,x\right)\\ a_{k}\left(u;t,x\right) & =\lim_{\Delta x_{k}\to 0}\frac{1}{\Delta x_{k}}{\tilde{M}}_{1}\left(u;t,x,\Delta x_{k}\right)\\ b_{k}\left(u;t,x\right) & =\lim_{\Delta x_{k}\to 0}\frac{1}{\Delta x_{k}}{\tilde{M}}_{2}\left(u;t,x,\Delta x_{k}\right)\\ k & =1,2,\cdots ,d \end{array}\right.$$

*here,*

(45)
$$\left\{\begin{array}{cc} \; & M_{n}\left(u;t,\Delta t,x\right)=E\left\{{\left[X(t+\Delta t,x)-X(t,x)\right]}^{n}|X\left(t,x\right)=u\right\}\\ \; & {\tilde{M}}_{n}\left(u;t,x,\Delta x_{k}\right)=E\left\{{\left[X\left(t,x+\Delta x_{k}e_{k}\right)-X\left(t,x\right)\right]}^{n}|X\left(t,x\right)=u\right\} \end{array}\right.$$

*are n-order conditional moments and $e_{k}=\left(0,0,\cdots ,1,\cdots ,0\right)\in R^{d}$ are standard orthogonal basis vectors, k=1,2,⋯,d.*


**Proof.** ∀Δt≠0, we can obtain the difference of density function in the time variable
(46)$$p\left(u;t+\Delta t,x\right)-p\left(u;t,x\right)=\sum_{n=1}^{+\infty }\frac{{\left(-1\right)}^{n}}{n!}\frac{\partial^{n}}{\partial u^{n}}\left[M_{n}\left(u;t,\Delta t,x\right)p\left(u;t,x\right)\right]$$
where
(47)$$M_{n}\left(u;t,\Delta t,x\right)=E\left\{{\left[X(t+\Delta t,x)-X(t,x)\right]}^{n}|X\left(t,x\right)=u\right\}$$
is the *n*-order conditional moment. Then, the partial derivative of the density function with respect to *t* is
(48)$$\frac{\partial }{\partial t}p\left(u;t,x\right)=\lim_{\Delta t\to 0}\frac{1}{\Delta t}\sum_{n=1}^{+\infty }\frac{{\left(-1\right)}^{n}}{n!}\frac{\partial^{n}}{\partial u^{n}}\left[M_{n}\left(u;t,\Delta t,x\right)p\left(u;t,x\right)\right]$$The Pawula theorem implies that if the Kramers–Moyal expansion stops after the second term, we obtain the Fokker–Planck equation about the time variable *t*
(49)$$\frac{\partial }{\partial t}p\left(u;t,x\right)=\frac{1}{2}\frac{\partial^{2}}{\partial u^{2}}\left[b_{0}\left(u;t,x\right)p\left(u,t,x\right)\right]-\frac{\partial }{\partial u}\left[a_{0}\left(u;t,x\right)p\left(u;t,x\right)\right]$$
where
(50)$$\left\{\begin{array}{cc} \; & a_{0}\left(u;t,x\right)=\lim_{\Delta t\to 0}\frac{1}{\Delta t}M_{1}\left(u;t,\Delta t,x\right)\\ \; & b_{0}\left(u;t,x\right)=\lim_{\Delta t\to 0}\frac{1}{\Delta t}M_{2}\left(u;t,\Delta t,x\right) \end{array}\right.$$Similarly, we may consider the increment $\Delta x_{k}$ of the spatial variable $x_{k}$, and we can obtain the Fokker–Planck equations about $x_{k}$ as
(51)$$\frac{\partial }{\partial x_{k}}p\left(u;t,x\right)=\frac{1}{2}\frac{\partial^{2}}{\partial u^{2}}\left[b_{k}\left(u;t,x\right)p\left(u,t,x\right)\right]-\frac{\partial }{\partial u}\left[a_{k}\left(u;t,x\right)p\left(u;t,x\right)\right]$$
where
(52)$$\left\{\begin{array}{cc} \; & a_{k}\left(u;t,x\right)=\lim_{\Delta x_{k}\to 0}\frac{1}{\Delta x_{k}}{\tilde{M}}_{1}\left(u;t,x,\Delta x_{k}\right)\\ \; & b_{k}\left(u;t,x\right)=\lim_{\Delta x_{k}\to 0}\frac{1}{\Delta x_{k}}{\tilde{M}}_{2}\left(u;t,x,\Delta x_{k}\right) \end{array}\right.$$
here,
(53)$${\tilde{M}}_{n}\left(u;t,x,\Delta x_{k}\right)=E\left\{{\left[X\left(t,x+\Delta x_{k}e_{k}\right)-X\left(t,x\right)\right]}^{n}|X\left(t,x\right)=u\right\}$$$e_{k}=\left(0,0,\cdots ,1,\cdots ,0\right)\in R^{d}$ are standard orthogonal basis vectors, k=1,2,⋯,d. □

The Fokker–Planck equations are partial differential equations that describe the probability density function of the space–time random field, similar to the classical Fokker–Planck equation. Solving a system of partial differential equations for general Fokker–Planck equations proves to be challenging. Fortunately, in Section 4 we present three distinct categories of space–time random fields in detail, along with their corresponding Fokker–Planck equations, and deduce their probability density functions.

Next, we examine the relationship between Jeffreys divergence and Fisher information in a single space–time random field when there are different time or spatial variables.

**Theorem** **2.**
*Suppose that p(u;t,x)>0 is a continuous differential density function of the space–time random field X(t,x), the partial derivatives $\partial_{u}p\left(u;t,x\right)$, $\partial_{t}p\left(u;t,x\right)$, $\partial_{x_{k}}p\left(u;t,x\right)$ are continuous bounded functions, and the integrals in the proof are well-defined, k=1,2,⋯,d, u∈R, $\left(t,x\right)\in R_{+}\times R^{d}$. Then, we have*

(54)
$$\left\{\begin{array}{cc} \lim_{\left|t-s\right|\to 0}\frac{JD\left(P(t,x),P(s,x)\right)}{{\left|t-s\right|}^{2}} & =FI_{1}^{\left(t\right)}\left(X(t,x)\right)\\ \lim_{\left|x_{k}-x_{k}^{\prime }\right|\to 0}\frac{JD\left(P\left(t,x\right),P\left(t,{\tilde{x}}^{\left(k\right)}\right)\right)}{|x_{k}-x_{k}^{\prime }{|}^{2}} & =FI_{1}^{\left(x_{k}\right)}\left(X(t,x)\right)\\ k & =1,2,\cdots ,d \end{array}\right.$$



**Proof.** For fixed x∈R, and ∀s≠t>0,
(55)$$\begin{array}{cc} JD\left(P(t,x),P(s,x)\right) & =KL\left(P(t,x)||P(s,x)\right)+KL\left(P(s,x)||P(t,x)\right)\\ \; & =\int_{R}\left[\log p(u;t,x)-\log p(u,s,x)\right]\left[p(u;t,x)-p(u,s,x)\right]du \end{array}$$
then we can obtain
(56)$$\begin{array}{cc} \; & \lim_{\left|t-s\right|\to 0}\frac{JD\left(P(t,x),P(s,x)\right)}{{\left|t-s\right|}^{2}}\\ = & \lim_{\left|t-s\right|\to 0}\int_{R}\frac{\log p(u;t,x)-\log p(u,s,x)}{t-s}\frac{p(u;t,x)-p(u,s,x)}{t-s}du \end{array}$$Notice that
(57)$$\left\{\begin{array}{cc} \lim_{\left|t-s\right|\to 0}\frac{\log p(u;t,x)-\log p(u,s,x)}{t-s} & =\partial_{t}\log p\left(u;t,x\right)\\ \lim_{\left|t-s\right|\to 0}\frac{p(u;t,x)-p(u,s,x)}{t-s} & =\partial_{t}p\left(u;t,x\right) \end{array}\right.$$
exist, and we obtain
(58)$$\begin{array}{cc} \; & \lim_{\left|t-s\right|\to 0}\frac{JD\left(P(t,x),P(s,x)\right)}{{\left|t-s\right|}^{2}}\\ = & \int_{R}\partial_{t}\log p\left(u;t,x\right)\partial_{t}p\left(u;t,x\right)du\\ = & \int_{R}\partial_{t}{\left[\log p(u;t,x)\right]}^{2}p\left(u;t,x\right)du\\ = & FI_{1}^{\left(t\right)}\left(X(t,x)\right) \end{array}$$Similarly, for fixed *t* and $\forall x_{k}\neq x_{k}^{\prime }$, we can obtain the identity on Jeffreys divergence and Fisher information for space coordinates
(59)$$\lim_{\left|x_{k}-x_{k}^{\prime }\right|\to 0}\frac{JD\left(P\left(t,x\right),P\left(t,{\tilde{x}}^{\left(k\right)}\right)\right)}{|x_{k}-x_{k}^{\prime }{|}^{2}}=FI_{1}^{\left(x_{k}\right)}\left(X(t,x)\right)$$
for k=1,2,⋯,d. □

Theorem 2 states that as the space–time variable difference approaches zero, the Fisher information of the space–time random field is the limit of the ratio of Jeffreys divergence at different locations to the square of space–time variable difference. It is noteworthy that Theorem 2 specifically addresses Jeffreys divergence only in cases where a single space–time random field is situated in distinct space–time positions, and where the difference between space–time variables approaches to 0. This ensures that Jeffreys divergence will not be infinite.

**Theorem** **3.**
*Suppose that p(u;t,x) and q(u;t,x) are continuous differentiable density functions of space–time random fields X(t,x) and Y(t,x) such that*

(60)
$$\left\{\begin{array}{cc} \; & \lim_{u\to \infty }\left\{\frac{1}{2}\partial_{u}\left[b_{k}^{\left(1\right)}\left(u;t,x\right)p\left(u;t,x\right)\right]-a_{k}^{\left(1\right)}\left(u;t,x\right)p\left(u;t,x\right)\right\}\left[\log\frac{p(u;t,x)}{q(u;t,x)}-\frac{q(u;t,x)}{p(u;t,x)}\right]=0\\ \; & \lim_{u\to \infty }\left\{\frac{1}{2}\partial_{u}\left[b_{k}^{\left(2\right)}\left(u;t,x\right)q\left(u;t,x\right)\right]-a_{k}^{\left(2\right)}\left(u;t,x\right)q\left(u;t,x\right)\right\}\left[\log\frac{q(u;t,x)}{p(u;t,x)}-\frac{p(u;t,x)}{q(u;t,x)}\right]=0 \end{array}\right.$$

*where $a_{k}$, $b_{k}$ are the forms in (Equation 44) and (Equation 45), and $\left(t,x\right)\in R_{+}\times R^{d}$, k=0,1,2,⋯,d. Then, the Jeffreys divergence $JD\left(P(t,x),Q(t,x)\right)$ satisfies generalized De Bruijn identities*

(61)
$$\left\{\begin{array}{cc} \frac{\partial }{\partial t}JD\left(P(t,x),Q(t,x)\right) & =-\frac{1}{2}FD_{\left(b_{0}^{\left(1\right)},b_{0}^{\left(2\right)}\right)}\left(P(t,x)∥Q(t,x)\right)-R_{0}\left(P(t,x)∥Q(t,x)\right)\\ \frac{\partial }{\partial x_{k}}JD\left(P(t,x),Q(t,x)\right) & =-\frac{1}{2}FD_{\left(b_{k}^{\left(1\right)},b_{k}^{\left(2\right)}\right)}\left(P(t,x)∥Q(t,x)\right)-R_{k}\left(P(t,x)∥Q(t,x)\right)\\ k & =1,2,\cdots ,d \end{array}\right.$$

*where*

(62)
$$\left\{\begin{array}{cc} R_{0}\left(P(t,x)∥Q(t,x)\right) & =\int_{R}\left[\frac{1}{2}\partial_{uu}^{2}\left(b_{0}^{\left(1\right)}-b_{0}^{\left(2\right)}\right)-\partial_{u}\left(a_{0}^{\left(1\right)}-a_{0}^{\left(2\right)}\right)\right]\left(p+q\right)du\\ R_{k}\left(P(t,x)∥Q(t,x)\right) & =\int_{R}\left[\frac{1}{2}\partial_{uu}^{2}\left(b_{k}^{\left(1\right)}-b_{k}^{\left(2\right)}\right)-\partial_{u}\left(a_{k}^{\left(1\right)}-a_{k}^{\left(2\right)}\right)\right]\left(p+q\right)du\\ k & =1,2,\cdots ,d \end{array}\right.$$

*here, we omit (u;t,x) in the integrals for convenience.*


**Proof.** By Definition 2, we have
(63)$$\begin{array}{cc} \; & JD\left(P(t,x),Q(t,x)\right)\\ = & KL(P(t,x)∥Q(t,x))+KL(Q(t,x)∥P(t,x))\\ = & \int_{R}p\log\frac{p}{q}du+\int_{R}q\log\frac{q}{p}du\\ = & \int_{R}\left[p\log\frac{p}{q}+q\log\frac{q}{p}\right]du \end{array}$$
where p:=p(u;t,x), q:=q(u;t,x) are density functions of X(t,x) and Y(t,x); here, we omit (u;t,x).Notice that
(64)$$\left\{\begin{array}{cc} \partial_{u}\left(\frac{p}{q}\right) & =\frac{1}{q}\left(\partial_{u}p-\frac{p}{q}\partial_{u}q\right)\\ \partial_{u}\left(\frac{q}{p}\right) & =\frac{1}{p}\left(\partial_{u}q-\frac{q}{p}\partial_{u}p\right) \end{array}\right.$$
i.e.,
(65)$$\left\{\begin{array}{cc} \frac{p}{q}\partial_{u}q & =\partial_{u}p-q\partial_{u}\left(\frac{p}{q}\right)\\ \frac{q}{p}\partial_{u}p & =\partial_{u}q-p\partial_{u}\left(\frac{q}{p}\right) \end{array}\right.$$
and
(66)$$\begin{array}{cc} \; & \left(\partial_{u}\log p-\partial_{u}\log q\right)\left(p+q\right)\\ = & \left(\frac{1}{p}\partial_{u}p-\frac{1}{q}\partial_{u}q\right)\left(p+q\right)\\ = & \partial_{u}p-\partial_{u}q+\frac{q}{p}\partial_{u}p-\frac{p}{q}\partial_{u}q\\ = & p\frac{q\partial_{u}p-p\partial_{u}q}{p^{2}}+q\frac{q\partial_{u}p-p\partial_{u}q}{q^{2}}\\ = & -\left[p\partial_{u}\left(\frac{q}{p}\right)-q\partial_{u}\left(\frac{p}{q}\right)\right] \end{array}$$
then,
(67)$$\begin{array}{cc} \; & \partial_{t}JD\left(P(t,x),Q(t,x)\right)\\ = & \int_{R}\left[\partial_{t}p\log\frac{p}{q}+q\partial_{t}\left(\frac{p}{q}\right)+\partial_{t}q\log\frac{q}{p}+p\partial_{t}\left(\frac{q}{p}\right)\right]du\\ = & \int_{R}\left[\partial_{t}p\log\frac{p}{q}+\partial_{t}p-\frac{p}{q}\partial_{t}q+\partial_{t}q\log\frac{q}{p}+\partial_{t}q-\frac{q}{p}\partial_{t}p\right]du\\ = & \int_{R}\left[\left(\log\frac{p}{q}-\frac{q}{p}\right)\partial_{t}p+\left(\log\frac{q}{p}-\frac{p}{q}\right)\partial_{t}q\right]du\\ = & \int_{R}\left(\log\frac{p}{q}-\frac{q}{p}\right)\left[\frac{1}{2}\partial_{uu}^{2}\left(b_{0}^{\left(1\right)}p\right)-\partial_{u}\left(a_{0}^{\left(1\right)}p\right)\right]du\\ \; & +\int_{R}\left(\log\frac{q}{p}-\frac{p}{q}\right)\left[\frac{1}{2}\partial_{uu}^{2}\left(b_{0}^{\left(2\right)}q\right)-\partial_{u}\left(a_{0}^{\left(2\right)}q\right)\right]du\\ = & -\int_{R}\left[\frac{1}{2}\partial_{u}\left(b_{0}^{\left(1\right)}p\right)-a_{0}^{\left(1\right)}p\right]\left[\frac{q}{p}\partial_{u}\left(\frac{p}{q}\right)-\partial_{u}\left(\frac{q}{p}\right)\right]du\\ \; & -\int_{R}\left[\frac{1}{2}\partial_{u}\left(b_{0}^{\left(2\right)}q\right)-a_{0}^{\left(2\right)}q\right]\left[\frac{p}{q}\partial_{u}\left(\frac{q}{p}\right)-\partial_{u}\left(\frac{p}{q}\right)\right]du\\ = & -\int_{R}\left[\frac{1}{2}\partial_{u}\left(b_{0}^{\left(1\right)}p\right)-a_{0}^{\left(1\right)}p\right]\frac{1}{p}\left[q\partial_{u}\left(\frac{p}{q}\right)-p\partial_{u}\left(\frac{q}{p}\right)\right]du\\ \; & -\int_{R}\left[\frac{1}{2}\partial_{u}\left(b_{0}^{\left(2\right)}q\right)-a_{0}^{\left(2\right)}q\right]\frac{1}{q}\left[p\partial_{u}\left(\frac{q}{p}\right)-q\partial_{u}\left(\frac{p}{q}\right)\right]du\\ = & -\int_{R}\left\{\left[\frac{1}{2}\partial_{u}\left(b_{0}^{\left(1\right)}p\right)-a_{0}^{\left(1\right)}p\right]\frac{1}{p}-\left[\frac{1}{2}\partial_{u}\left(b_{0}^{\left(2\right)}q\right)-a_{0}^{\left(2\right)}q\right]\frac{1}{q}\right\}\\ \; & \times \left(\partial_{u}\log p-\partial_{u}\log q\right)\left(p+q\right)du\\ = & -\int_{R}\left[\frac{1}{2}b_{0}^{\left(1\right)}\partial_{u}\log p-\frac{1}{2}b_{0}^{\left(2\right)}\partial_{u}\log q+\frac{1}{2}\partial_{u}\left(b_{0}^{\left(1\right)}-b_{0}^{\left(2\right)}\right)-\left(a_{0}^{\left(1\right)}-a_{0}^{\left(2\right)}\right)\right]\\ \; & \times \left(\partial_{u}\log p-\partial_{u}\log q\right)\left(p+q\right)du\\ = & -\int_{R}\left[\frac{1}{2}b_{0}^{\left(1\right)}\partial_{u}\log p-\frac{1}{2}b_{0}^{\left(2\right)}\partial_{u}\log q\right]\left(\partial_{u}\log p-\partial_{u}\log q\right)\left(p+q\right)du\\ \; & -\int_{R}\left[\frac{1}{2}\partial_{u}\left(b_{0}^{\left(1\right)}-b_{0}^{\left(2\right)}\right)-\left(a_{0}^{\left(1\right)}-a_{0}^{\left(2\right)}\right)\right]\left(\partial_{u}\log p-\partial_{u}\log q\right)\left(p+q\right)du\\ = & -\frac{1}{2}FD_{\left(b_{0}^{\left(1\right)},b_{0}^{\left(2\right)}\right)}\left(P(t,x)∥Q(t,x)\right)-R_{0}\left(P(t,x)∥Q(t,x)\right) \end{array}$$
where
(68)$$\begin{array}{cc} \; & FD_{\left(b_{0}^{\left(1\right)},b_{0}^{\left(2\right)}\right)}\left(P(t,x)∥Q(t,x)\right)\\ = & \int_{R}\left[b_{0}^{\left(1\right)}\partial_{u}\log p-b_{0}^{\left(2\right)}\partial_{u}\log q\right]\left(\partial_{u}\log p-\partial_{u}\log q\right)\left(p+q\right)du \end{array}$$
and
(69)$$R_{0}\left(P(t,x)∥Q(t,x)\right)=\int_{R}\left[\frac{1}{2}\partial_{uu}^{2}\left(b_{0}^{\left(1\right)}-b_{0}^{\left(2\right)}\right)-\partial_{u}\left(a_{0}^{\left(1\right)}-a_{0}^{\left(2\right)}\right)\right]\left(p+q\right)du$$Similarly, for k=1,2,⋯,d, we can obtain the generalized De Bruijn identities about the spatial variable $x_{k}$
(70)$$\frac{\partial }{\partial x_{k}}JD\left(P(t,x),Q(t,x)\right)=-\frac{1}{2}FD_{\left(b_{k}^{\left(1\right)},b_{k}^{\left(2\right)}\right)}\left(P(t,x)∥Q(t,x)\right)-R_{k}\left(P\left(t,x\right),Q\left(t,x\right)\right)$$
where
(71)$$R_{k}\left(P(t,x)∥Q(t,x)\right)=\int_{R}\left[\frac{1}{2}\partial_{uu}^{2}\left(b_{k}^{\left(1\right)}-b_{k}^{\left(2\right)}\right)-\partial_{u}\left(a_{k}^{\left(1\right)}-a_{k}^{\left(2\right)}\right)\right]\left(p+q\right)du$$
then we obtain the conclusion. □

Unlike Theorem 3, Theorem 4 focuses on the Jeffreys divergence between two separate space–time random fields X(t,x) and Y(t,x), both at the same position (t,x), and establishes the identities of the connection between the Jeffreys divergence and the Fisher divergence of X(t,x) and Y(t,x). This is known as the De Bruijn identities. To prevent Jeffreys divergence from becoming infinite, it is necessary for the difference between the probability density functions of X(t,x) and Y(t,x) to be small. In Section 4, we obtain Jeffreys divergence and Fisher divergence using the same type of Fokker–Planck equations but with different parameters. This allows for the selection of only the appropriate parameters.

## 4. Three Fokker–Planck Random Fields and Their Corresponding Information Measures

In this section, we present three types of Fokker–Planck equations and derive their corresponding density functions and information measures, which are Jeffreys divergence, generalized Fisher information, and Fisher divergence. With these quantities, the results corresponding to the applications of Theorems 2 and 3 are obtained. On the one hand, we calculate the ratio of Jeffreys divergence to the square of space–time variation on the identical Fokker–Planck space–time random field at various space–time points, in comparison to generalized Fisher information. On the other hand, we derive the De Burijn identities for Jeffreys divergence and generalized Fisher divergence from Fokker–Planck equations on a single space–time random field at the corresponding space–time location, under same type but with different parameters.

First, we present a theorem regarding simple type Fokker–Planck equations of the random field.

**Theorem** **4.**
*Suppose the functions in the Fokker–Planck Equations (Equation 43) for the continuous random field X(t,x) are formulated as follows:*

(72)
$$\left\{\begin{array}{cc} a_{0}\left(u;t,x\right) & =a_{0}\left(t,x\right)\\ b_{0}\left(u;t,x\right) & =b_{0}\left(t,x\right)>0\\ a_{k}\left(u;t,x\right) & =a_{k}\left(t,x\right)\\ b_{k}\left(u;t,x\right) & =b_{k}\left(t,x\right)>0\\ k & =1,2,\cdots ,d \end{array}\right.$$

*where $a_{0}$, $a_{k}$, $b_{0}$, and $b_{k}$ are continuously differentiable functions independent of u and two continuously differentiable functions, α(t,x) and β(t,x), exist such that*

(73)
$$\left\{\begin{array}{cc} \; & d\alpha \left(t,x\right)=a_{0}dt+a_{1}dx_{1}+\cdots +a_{d}dx_{d}\\ \; & d\beta \left(t,x\right)=b_{0}dt+b_{1}dx_{1}+\cdots +b_{d}dx_{d} \end{array}\right.$$

*the initial density function is $p\left(u;t,x\right)=\delta \left[u-u_{0}\left(x\right)\right]$ as prod(t,x)=0; then, the density function of X(t,x) is presented as follows:*

(74)
$$p\left(u;t,x\right)=\frac{1}{\sqrt{2\pi \beta (t,x)}}e^{-\frac{{\left[u-u_{0}\left(x\right)-\alpha \left(t,x\right)\right]}^{2}}{2\beta (t,x)}}$$



**Proof.** It can be easily inferred that the Fokker–Planck equations are simple parabolic equations, and their solution can be obtained through Fourier transform
(75)$$\left\{\begin{array}{cc} p(u;t,x) & =\frac{1}{\sqrt{2\pi \int_{0}^{t}b_{0}\left(s,x\right)ds}}e^{-\frac{{\left[u-u_{0}\left(x\right)-\int_{0}^{t}a_{0}\left(s,x\right)ds\right]}^{2}}{2\int_{0}^{t}b_{0}\left(s,x\right)ds}}\\ p(u;t,x) & =\frac{1}{\sqrt{2\pi \int_{0}^{x_{k}}b_{k}\left(t,x\right)dx_{k}}}e^{-\frac{{\left[u-u_{0}\left(x\right)-\int_{0}^{x_{k}}a_{k}\left(t,x\right)dx_{k}\right]}^{2}}{2\int_{0}^{x_{k}}b_{k}\left(t,x\right)dx_{k}}} \end{array}\right.$$Recall that there are two functions α(t,x) and β(t,x) such that
(76)$$\left\{\begin{array}{cc} \; & d\alpha \left(t,x\right)=a_{0}\left(t,x\right)dt+a_{1}\left(t,x\right)dx_{1}+\cdots +a_{d}\left(t,x\right)dx_{d}\\ \; & d\beta \left(t,x\right)=b_{0}\left(t,x\right)dt+b_{1}\left(t,x\right)dx_{1}+\cdots +b_{d}\left(t,x\right)dx_{d} \end{array}\right.$$
we can obtain the probability density function
(77)$$p\left(u;t,x\right)=\frac{1}{\sqrt{2\pi \beta (t,x)}}e^{-\frac{{\left[u-u_{0}\left(x\right)-\alpha \left(t,x\right)\right]}^{2}}{2\beta (t,x)}}$$□

Actually, numerous examples exist in which the Fokker–Planck equations comply with Theorem 4. Let B(t,x) be the (1+d,1) Brownian sheet [52,53], that is, a centered continuous Gaussian process that is indexed by (1+d) real, positive parameters and takes its values in R. Moreover, its covariance structure is given by
(78)$$E\left[B(t,x)B(s,y)\right]=\left(t\wedge s\right)\prod_{k=1}^{d}\left(x_{k}\wedge y_{k}\right)$$
for $\left(t,x_{1},x_{2},\cdots ,x_{d}\right)$, $\left(s,y_{1},y_{2},\cdots ,y_{d}\right)\in R_{+}\times R_{+}^{d}$, where $\left(\cdot \wedge \cdot \right)$ represents the minimum of two numbers. We can easily obtain
(79)$$E\left[B^{2}\left(t,x\right)\right]=\mathrm{prod}\left(t,x\right)$$
where $\mathrm{prod}\left(t,x\right)=tx_{1}x_{2}\cdots x_{d}$ is the coordinate product of (t,x) and the density function is
(80)$$p^{\left(1\right)}\left(u;t,x\right)=\frac{1}{\sqrt{2\pi \mathrm{prod}(t,x)}}e^{-\frac{u^{2}}{2\mathrm{prod}(t,x)}}$$

Moreover, the Fokker–Planck equations of Brownian sheet are
(81)$$\left\{\begin{array}{cc} \frac{\partial }{\partial t}p^{\left(1\right)}\left(u;t,x\right) & =\frac{\mathrm{prod}(x)}{2}\frac{\partial^{2}}{\partial u^{2}}p^{\left(1\right)}\left(u,t,x\right)\\ \frac{\partial }{\partial x_{k}}p^{\left(1\right)}\left(u;t,x\right) & =\frac{\mathrm{prod}(t,x)}{2x_{k}}\frac{\partial^{2}}{\partial u^{2}}p^{\left(1\right)}\left(u,t,x\right)\\ k & =1,2,\cdots ,d \end{array}\right.$$
with the initial condition p(u;t,x)=δ(u) as prod(t,x)=0.

Following the concept of constructing a Brownian bridge on Brownian motion [53], we refer to
(82)$$B^{*}\left(t,x\right)=B\left(t,x\right)-\mathrm{prod}\left(t,x\right)B\left(1,1,\cdots ,1\right)$$
as a Brownian sheet bridge on the cube $\left(t,x\right)\in \left[0,1\right]\times {\left[0,1\right]}^{d}$, where B(t,x) represents the Brownian sheet. Obviously, $B^{*}\left(t,x\right)$ is Gaussian, and $E\left[B^{*}\left(t,x\right)\right]=0$ and the covariance structure are
(83)$$E\left[B^{*}\left(t,x\right)B^{*}\left(s,y\right)\right]=E\left[B(t,x)B(s,y)\right]-\mathrm{prod}\left(t,x\right)\mathrm{prod}\left(s,y\right)$$
we can obtain
(84)$$E\left[B^{2}\left(t,x\right)\right]=\mathrm{prod}\left(t,x\right)\left[1-\mathrm{prod}(t,x)\right]$$
and the density function of $B^{*}\left(t,x\right)$ is
(85)$$p^{\left(2\right)}\left(u;t,x\right)=\frac{1}{\sqrt{2\pi \mathrm{prod}\left(t,x\right)\left[1-\mathrm{prod}(t,x)\right]}}e^{-\frac{u^{2}}{2\mathrm{prod}\left(t,x\right)\left[1-\mathrm{prod}(t,x)\right]}}$$

In addition to this, the Fokker–Planck equations of Brownian sheet bridge are
(86)$$\left\{\begin{array}{cc} \frac{\partial }{\partial t}p^{\left(2\right)}\left(u;t,x\right) & =\frac{\mathrm{prod}\left(x\right)\left[1-2\mathrm{prod}(t,x)\right]}{2}\frac{\partial^{2}}{\partial u^{2}}p^{\left(2\right)}\left(u,t,x\right)\\ \frac{\partial }{\partial x_{k}}p^{\left(2\right)}\left(u;t,x\right) & =\frac{\mathrm{prod}(t,x)}{2x_{k}}\left[1-2\mathrm{prod}(t,x)\right]\frac{\partial^{2}}{\partial u^{2}}p^{\left(2\right)}\left(u,t,x\right)\\ k & =1,2,\cdots ,d \end{array}\right.$$
with the initial condition p(u;t,x)=δ(u) as prod(t,x)=0, and we obtain the solution (Equation 85).

Combining two probability density functions (Equation 80) and (Equation 85) yields their respective Jeffreys divergence and generalized De Burijn identities. The Jeffreys divergence of (Equation 74) can be obtained at various space–time points as
(87)$$JD\left(P(t,x),P(s,y)\right)=\frac{{\left[\alpha (t,x)-\alpha (s,y)\right]}^{2}+\beta \left(s,y\right)}{2\beta (t,x)}+\frac{{\left[\alpha (t,x)-\alpha (s,y)\right]}^{2}+\beta \left(t,x\right)}{2\beta (s,y)}-1$$
and the Fisher divergence between $P^{\left(1\right)}$ and $P^{\left(2\right)}$ at the identical space–time point represents
(88)$$\begin{array}{cc} \; & FD_{\left(b_{k}^{\left(1\right)},b_{k}^{\left(2\right)}\right)}\left(P^{\left(1\right)}\left(t,x\right)∥P^{\left(2\right)}\left(t,x\right)\right)\\ = & \frac{1}{\beta_{1}^{2}\left(t,x\right)\beta_{2}^{2}\left(t,x\right)}\{{\left[\alpha_{1}\left(t,x\right)-\alpha_{2}\left(t,x\right)\right]}^{2}\left[b_{k}^{\left(2\right)}\beta_{1}^{2}\left(t,x\right)+b_{k}^{\left(1\right)}\beta_{2}^{2}\left(t,x\right)\right]\\ \; & +\left[\beta_{1}\left(t,x\right)-\beta_{2}\left(t,x\right)\right]\left[\beta_{1}\left(t,x\right)+\beta_{2}\left(t,x\right)\right]\left[b_{k}^{\left(2\right)}\beta_{1}\left(t,x\right)-b_{k}^{\left(1\right)}\beta_{2}\left(t,x\right)\right]\} \end{array}$$
where k=0,1,⋯,d.

Bring the density function of Brownian sheet into Equation (Equation 87); we can easily obtain the Jeffreys divergence of the Brownian sheet at various space–time points as
(89)$$JD\left(P^{\left(1\right)}\left(t,x\right),P^{\left(1\right)}\left(s,y\right)\right)=\frac{\mathrm{prod}(s,y)}{2\mathrm{prod}(t,x)}+\frac{\mathrm{prod}(t,x)}{2\mathrm{prod}(s,y)}-1$$
and the generalized Fisher information on space–time variables is as follows:(90)$$\left\{\begin{array}{cc} FI_{1}^{\left(t\right)}\left(P^{\left(1\right)}\left(t,x\right)\right) & =\frac{1}{2t^{2}}\\ FI_{1}^{\left(x_{k}\right)}\left(P^{\left(1\right)}\left(t,x\right)\right) & =\frac{1}{2x_{k}^{2}} \end{array}\right.$$k=1,2,⋯,d.

Then, we can obtain quotients of the squared difference between Jeffreys divergence and space–time variables
(91)$$\left\{\begin{array}{cc} \; & \frac{JD\left(P^{\left(1\right)}\left(t,x\right),P^{\left(1\right)}\left(s,x\right)\right)}{{\left|t-s\right|}^{2}}=\frac{1}{2st}\\ \; & \frac{JD\left(P^{\left(1\right)}\left(t,x\right),P^{\left(1\right)}\left(t,{\tilde{x}}^{\left(k\right)}\right)\right)}{|x_{k}-x_{k}^{\prime }{|}^{2}}=\frac{1}{2x_{k}x_{k}^{\prime }} \end{array}\right.$$
and then we can obtain the relation between quotients and generalized Fisher information
(92)$$\left\{\begin{array}{cc} \; & \frac{JD\left(P^{\left(1\right)}\left(t,x\right),P^{\left(1\right)}\left(s,x\right)\right)}{{\left|t-s\right|}^{2}FI_{1}^{\left(t\right)}\left(P^{\left(1\right)}\left(t,x\right)\right)}=\frac{t}{s}\\ \; & \frac{JD\left(P^{\left(1\right)}\left(t,x\right),P^{\left(1\right)}\left(t,{\tilde{x}}^{\left(k\right)}\right)\right)}{|x_{k}-x_{k}^{\prime }{|}^{2}FI_{1}^{\left(x_{k}\right)}\left(P^{\left(1\right)}\left(t,x\right)\right)}=\frac{x_{k}}{x_{k}^{\prime }} \end{array}\right.$$
for k=1,2,⋯,d. If we consider the approximation of spacetime points (t,x) and (s,y), the final result (Equation 92) satisfies the conclusion of Theorem 2.

Similarly, we can obtain the Jeffreys divergence of Brownian sheet bridge at different space–time points
(93)$$JD\left(P^{\left(2\right)}\left(t,x\right),P^{\left(2\right)}\left(s,y\right)\right)=\frac{\mathrm{prod}\left(s,y\right)\left[1-\mathrm{prod}(s,y)\right]}{2\mathrm{prod}\left(t,x\right)\left[1-\mathrm{prod}(t,x)\right]}+\frac{\mathrm{prod}\left(t,x\right)\left[1-\mathrm{prod}(t,x)\right]}{2\mathrm{prod}\left(s,y\right)\left[1-\mathrm{prod}(s,y)\right]}-1$$
and the generalized Fisher information on space–time variables
(94)$$\left\{\begin{array}{cc} FI_{1}^{\left(t\right)}\left(P^{\left(2\right)}\left(t,x\right)\right) & =\frac{{\left[1-2\mathrm{prod}(t,x)\right]}^{2}}{2t^{2}{\left[1-\mathrm{prod}(t,x)\right]}^{2}}\\ FI_{1}^{\left(x_{k}\right)}\left(P^{\left(2\right)}\left(t,x\right)\right) & =\frac{{\left[1-2\mathrm{prod}(t,x)\right]}^{2}}{2x_{k}^{2}{\left[1-\mathrm{prod}(t,x)\right]}^{2}} \end{array}\right.$$k=1,2,⋯,d. Further, we can easily get the quotients of the squared difference between Jeffreys divergence and space–time variables
(95)$$\left\{\begin{array}{cc} \; & \frac{JD\left(P^{\left(2\right)}\left(t,x\right),P^{\left(2\right)}\left(s,x\right)\right)}{{\left|t-s\right|}^{2}}=\frac{{\left[1-\mathrm{prod}(x)(s+t)\right]}^{2}}{2st\left[1-\mathrm{prod}(s,x)\right]\left[1-\mathrm{prod}(t,x)\right]}\\ \; & \frac{JD\left(P^{\left(2\right)}\left(t,x\right),P^{\left(2\right)}\left(t,{\tilde{x}}^{\left(k\right)}\right)\right)}{|x_{k}-x_{k}^{\prime }{|}^{2}}=\frac{1}{2x_{k}x_{k}^{\prime }\left[1-\mathrm{prod}(t,x)\right]\left[1-\mathrm{prod}(t,{\tilde{x}}^{\left(k\right)})\right]}{\left[1-\frac{\mathrm{prod}(t,x)}{x_{k}}\left(x_{k}+x_{k}^{\prime }\right)\right]}^{2} \end{array}\right.$$
and then we can obtain the relation between quotients and generalized Fisher information
(96)$$\left\{\begin{array}{cc} \frac{JD\left(P^{\left(2\right)}\left(t,x\right),P^{\left(2\right)}\left(s,x\right)\right)}{{\left|t-s\right|}^{2}FI_{1}^{\left(t\right)}\left(P^{\left(2\right)}\left(t,x\right)\right)}= & \frac{t\left[1-\mathrm{prod}(t,x)\right]{\left[1-\mathrm{prod}(x)(s+t)\right]}^{2}}{s\left[1-\mathrm{prod}(s,x)\right]{\left[1-2\mathrm{prod}(t,x)\right]}^{2}}\\ \frac{JD\left(P^{\left(2\right)}\left(t,x\right),P^{\left(2\right)}\left(t,{\tilde{x}}^{\left(k\right)}\right)\right)}{|x_{k}-x_{k}^{\prime }{|}^{2}FI_{1}^{\left(x_{k}\right)}\left(P^{\left(2\right)}\left(t,x\right)\right)}= & \frac{x_{k}\left[1-\mathrm{prod}(t,x)\right]}{x_{k}^{\prime }\left[1-\mathrm{prod}(t,{\tilde{x}}^{\left(k\right)})\right]{\left[1-2\mathrm{prod}(t,x)\right]}^{2}}{\left[1-\frac{\mathrm{prod}(t,x)}{x_{k}}\left(x_{k}+x_{k}^{\prime }\right)\right]}^{2} \end{array}\right.$$
for k=1,2,⋯,d. Without loss of generality, the result (Equation 96) also satisfies Theorem 2.

Next, we evaluate the Jeffreys divergence between the density functions (Equation 80) and (Equation 85) for the same space–time points. It should be noted that the Brownian sheet bridge density function is defined on a bounded domain; therefore, we limit our analysis to the space–time region of $\left(t,x\right)\in \left[0,1\right]\times {\left[0,1\right]}^{d}$.

The Jeffreys divergence between $P^{\left(1\right)}$ and $P^{\left(2\right)}$ can be easily obtained as
(97)$$JD\left(P^{\left(1\right)}\left(t,x\right),P^{\left(2\right)}\left(t,x\right)\right)=\frac{1-\mathrm{prod}(t,x)}{2}+\frac{1}{2\left[1-\mathrm{prod}(t,x)\right]}-1$$
and the Fisher divergence as shown in (Equation 88) is given by
(98)$$\left\{\begin{array}{cc} \; & FD_{\left(b_{0}^{\left(1\right)},b_{0}^{\left(2\right)}\right)}\left(P^{\left(1\right)}\left(t,x\right)∥P^{\left(2\right)}\left(t,x\right)\right)=\mathrm{prod}\left(x\right)-\frac{\mathrm{prod}(x)}{{\left[1-\mathrm{prod}(t,x)\right]}^{2}}\\ \; & FD_{\left(b_{k}^{\left(1\right)},b_{k}^{\left(2\right)}\right)}\left(P^{\left(1\right)}\left(t,x\right)∥P^{\left(2\right)}\left(t,x\right)\right)=\frac{\mathrm{prod}(t,x)}{x_{k}}-\frac{\mathrm{prod}(t,x)}{x_{k}{\left[1-\mathrm{prod}(t,x)\right]}^{2}} \end{array}\right.$$
with the remainder terms
(99)$$R_{0}\left(P^{\left(1\right)}\left(t,x\right)∥P^{\left(2\right)}\left(t,x\right)\right)=R_{k}\left(P^{\left(1\right)}\left(t,x\right)∥P^{\left(2\right)}\left(t,x\right)\right)=0$$
for k=1,2,⋯,d. Furthermore, we can obtain the generalized De Bruijn identities
(100)$$\left\{\begin{array}{cc} \frac{\partial }{\partial t}JD\left(P^{\left(1\right)}\left(t,x\right),P^{\left(2\right)}\left(t,x\right)\right) & =-\frac{1}{2}FD_{\left(b_{0}^{\left(1\right)},b_{0}^{\left(2\right)}\right)}\left(P^{\left(1\right)}\left(t,x\right)∥P^{\left(2\right)}\left(t,x\right)\right)\\ \frac{\partial }{\partial x_{k}}JD\left(P^{\left(1\right)}\left(t,x\right),P^{\left(2\right)}\left(t,x\right)\right) & =-\frac{1}{2}FD_{\left(b_{k}^{\left(1\right)},b_{k}^{\left(2\right)}\right)}\left(P^{\left(1\right)}\left(t,x\right)∥P^{\left(2\right)}\left(t,x\right)\right)\\ k & =1,2,\cdots ,d \end{array}\right.$$

Next, we present two categories of significant Fokker–Planck equations and provide pertinent illustrations for computing Jefferys divergence, Fisher information, and Fisher divergence.

**Theorem** **5.**
*Suppose the functions in the Fokker–Planck Equations (Equation 43) for the continuous random field X(t,x) are formulated as follows:*

(101)
$$\left\{\begin{array}{cc} a_{k}\left(u;t,x\right) & \equiv 0\\ b_{k}\left(u;t,x\right) & =b_{k}\left(t,x\right)u^{2}>0\\ k & =0,1,2,\cdots ,d \end{array}\right.$$

*where $b_{k}$ are continuously differentiable functions independent of u and a continuously differentiable function β(t,x) exists, such that*

(102)
$$d\beta \left(t,x\right)=b_{0}\left(t,x\right)dt+b_{1}\left(t,x\right)dx_{1}+\cdots +b_{d}\left(t,x\right)dx_{d}$$

*the initial value X(t,x)=1 as prod(t,x)=0 and the initial density function is p(u;t,x)=δ(u−1) as prod(t,x)=0. Then, the density function is*

(103)
$$p\left(u;t,x\right)=\frac{e^{\beta (t,x)}}{\sqrt{2\pi \beta (t,x)}}e^{-\frac{{\left[\log u+\frac{3}{2}\beta \left(t,x\right)\right]}^{2}}{2\beta (t,x)}}$$



**Proof.** Depending on the conditions, it is easy to obtain the Fokker–Planck equations as
(104)$$\left\{\begin{array}{cc} \frac{\partial }{\partial t}p\left(u;t,x\right) & =\frac{b_{0}\left(t,x\right)}{2}u^{2}\frac{\partial^{2}p\left(u;t,x\right)}{\partial u^{2}}+2b_{0}\left(t,x\right)u\frac{\partial }{\partial u}p\left(u;t,x\right)+b_{0}\left(t,x\right)p\left(u;t,x\right)\\ \frac{\partial }{\partial x_{k}}p\left(u;t,x\right) & =\frac{b_{k}\left(t,x\right)}{2}u^{2}\frac{\partial^{2}p\left(u;t,x\right)}{\partial u^{2}}+2b_{k}\left(t,x\right)u\frac{\partial }{\partial u}p\left(u;t,x\right)+b_{k}\left(t,x\right)p\left(u;t,x\right)\\ k & =1,2,\cdots ,d \end{array}\right.$$Take the transformation v=logu or $u=e^{v}$ and note $\tilde{p}\left(v;t,x\right)=p\left(u\left(v\right);t,x\right)$; we can obtain
(105)$$\left\{\begin{array}{cc} \frac{\partial }{\partial t}\tilde{p}\left(v;t,x\right) & =\frac{b_{0}\left(t,x\right)}{2}\frac{\partial^{2}}{\partial v^{2}}\tilde{p}\left(v;t,x\right)+\frac{3b_{0}\left(t,x\right)}{2}\frac{\partial }{\partial v}\tilde{p}\left(v;t,x\right)+b_{0}\left(t,x\right)\tilde{p}\left(v;t,x\right)\\ \frac{\partial }{\partial x_{k}}\tilde{p}\left(v;t,x\right) & =\frac{b_{k}\left(t,x\right)}{2}\frac{\partial^{2}}{\partial v^{2}}\tilde{p}\left(v;t,x\right)+\frac{3b_{k}\left(t,x\right)}{2}\frac{\partial }{\partial v}\tilde{p}\left(v;t,x\right)+b_{k}\left(t,x\right)\tilde{p}\left(v;t,x\right)\\ k & =1,2,\cdots ,d \end{array}\right.$$
with the solution
(106)$$\left\{\begin{array}{cc} \tilde{p}\left(v;t,x\right) & =\frac{e^{\int_{0}^{t}b_{0}\left(s,x\right)ds}}{\sqrt{2\pi \int_{0}^{t}b_{0}\left(s,x\right)ds}}e^{-\frac{{\left[v+\frac{3}{2}\int_{0}^{t}b_{0}\left(s,x\right)ds\right]}^{2}}{2\int_{0}^{t}b_{0}\left(s,x\right)ds}}\\ \tilde{p}\left(v;t,x\right) & =\frac{e^{\int_{0}^{x_{k}}b_{k}\left(t,x\right)dx_{k}}}{\sqrt{2\pi \int_{0}^{x_{k}}b_{k}\left(t,x\right)dx_{k}}}e^{-\frac{{\left[v+\frac{3}{2}\int_{0}^{x_{k}}b_{k}\left(t,x\right)dx_{k}\right]}^{2}}{2\int_{0}^{x_{k}}b_{k}\left(t,x\right)dx_{k}}}\\ k & =1,2,\cdots ,d \end{array}\right.$$
then,
(107)$$\left\{\begin{array}{cc} p(u;t,x) & =\frac{e^{\int_{0}^{t}b_{0}\left(s,x\right)ds}}{\sqrt{2\pi \int_{0}^{t}b_{0}\left(s,x\right)ds}}e^{-\frac{{\left[\log u+\frac{3}{2}\int_{0}^{t}b_{0}\left(s,x\right)ds\right]}^{2}}{2\int_{0}^{t}b_{0}\left(s,x\right)ds}}\\ p(u;t,x) & =\frac{e^{\int_{0}^{x_{k}}b_{k}\left(t,x\right)dx_{k}}}{\sqrt{2\pi \int_{0}^{x_{k}}b_{k}\left(t,x\right)dx_{k}}}e^{-\frac{{\left[\log u+\frac{3}{2}\int_{0}^{x_{k}}b_{k}\left(t,x\right)dx_{k}\right]}^{2}}{2\int_{0}^{x_{k}}b_{k}\left(t,x\right)dx_{k}}}\\ k & =1,2,\cdots ,d \end{array}\right.$$Recall that a continuously differential function β(t,x) exists such that
(108)$$d\beta \left(t,x\right)=b_{0}\left(t,x\right)dt+b_{1}\left(t,x\right)dx_{1}+\cdots +b_{d}\left(t,x\right)dx_{d}$$
this enables the derivation of the probability density
(109)$$p\left(u;t,x\right)=\frac{e^{\beta (t,x)}}{\sqrt{2\pi \beta (t,x)}}e^{-\frac{{\left[\log u+\frac{3}{2}\beta \left(t,x\right)\right]}^{2}}{2\beta (t,x)}}$$□

**Remark** **1.***In the stochastic process theory, a correlation exists between the Fokker–Planck equation and the Itô process. Specifically, if the Itô process is*(110)$$dX_{t}=\mu \left(X_{t},t\right)dt+\sigma \left(X_{t},t\right)dB_{t}$$*then the corresponding Fokker–Planck equation can be obtained as*(111)$$\frac{\partial }{\partial t}p\left(u,t\right)=\frac{1}{2}\frac{\partial^{2}}{\partial u^{2}}\left[\sigma^{2}\left(u,t\right)p\left(u,t\right)\right]-\frac{\partial }{\partial u}\left[\mu (u,t)p(u,t)\right]$$*where μ and σ represent drift and diffusion, $B_{t}$ is the standard Brownian motion, or*(112)$$\frac{dX_{t}}{dt}=\mu \left(X_{t},t\right)+\sigma \left(X_{t},t\right)W_{t}$$*where $W_{t}=\frac{dB_{t}}{dt}$ represents the white noise. Actually, if we consider the Itô processes corresponding to Fokker–Planck equations from Theorem 5, we can obtain*(113)$$\left\{\begin{array}{cc} \partial_{t}X\left(t,x\right) & =\sqrt{b_{0}\left(t,x\right)}X\left(t,x\right)W_{t}\\ \partial_{x_{k}}X\left(t,x\right) & =\sqrt{b_{k}\left(t,x\right)}X\left(t,x\right)W_{k} \end{array}\right.$$*where $W_{k}$ represents the space white noise with respect to $x_{k}$, k=1,2,⋯,d. Further, we can also write Equation (Equation 113) in vector form*(114)∇X(t,x)=γ(t,x)X(t,x)⊙W(t,x)*where*(115)$$\left\{\begin{array}{cc} \gamma (t,x) & =\left[\sqrt{b_{0}\left(t,x\right)},\sqrt{b_{1}\left(t,x\right)},\cdots ,\sqrt{b_{d}\left(t,x\right)}\right]\\ W(t,x) & =\left[W_{t},W_{1},\cdots ,W_{d}\right] \end{array}\right.$$∇ *represents the gradient operator and* ⊙ *represents element by element multiplication. Notice that each equation in Equation (Equation 113) is similar in form to the geometric Brownian motion in the theory of stochastic processes. Similarly, we can call the space–time random field that satisfies Equation (Equation 113) a geometric Brownian filed.*

If we consider different $\beta_{3}\left(t,x\right)$ and $\beta_{4}\left(t,x\right)$ in density function (Equation 103), we can obtain density functions $p^{\left(3\right)}\left(u;t,x\right)$ and $p^{\left(4\right)}\left(u;t,x\right)$; then, we can obtain the Jeffreys divergence
(116)$$JD\left(P^{\left(3\right)}\left(t,x\right),P^{\left(3\right)}\left(s,y\right)\right)=\frac{\beta_{3}\left(t,x\right)+\beta_{3}\left(s,t\right)+4}{8\beta_{3}\left(t,x\right)\beta_{3}\left(s,y\right)}{\left[\beta_{3}\left(t,x\right)-\beta_{3}\left(s,y\right)\right]}^{2}$$
and generalized Fisher information
(117)$$\left\{\begin{array}{cc} FI_{1}^{\left(t\right)}\left(P^{\left(3\right)}\left(t,x\right)\right) & =\frac{\beta_{3}\left(t,x\right)+2}{4\beta_{3}^{2}\left(t,x\right)}{\left[b_{0}^{\left(3\right)}\left(t,x\right)\right]}^{2}\\ FI_{1}^{\left(x_{k}\right)}\left(P^{\left(3\right)}\left(t,x\right)\right) & =\frac{\beta_{3}\left(t,x\right)+2}{4\beta_{3}^{2}\left(t,x\right)}{\left[b_{k}^{\left(3\right)}\left(t,x\right)\right]}^{2} \end{array}\right.$$
and then the quotients
(118)$$\left\{\begin{array}{cc} \frac{JD\left(P^{\left(3\right)}\left(t,x\right),P^{\left(3\right)}\left(s,x\right)\right)}{{\left|t-s\right|}^{2}} & =\frac{\beta_{3}\left(t,x\right)+\beta_{3}\left(s,x\right)+4}{8\beta_{3}\left(t,x\right)\beta_{3}\left(s,y\right)}{\left[\frac{\beta_{3}\left(t,x\right)-\beta_{3}\left(s,y\right)}{t-s}\right]}^{2}\\ \frac{JD\left(P^{\left(3\right)}\left(t,x\right),P^{\left(3\right)}\left(t,{\tilde{x}}^{\left(k\right)}\right)\right)}{|x_{k}-x_{k}^{\prime }{|}^{2}} & =\frac{\beta_{3}\left(t,x\right)+\beta_{3}\left(t,{\tilde{x}}^{\left(k\right)}\right)+4}{8\beta_{3}\left(t,x\right)\beta_{3}\left(t,{\tilde{x}}^{\left(k\right)}\right)}{\left[\frac{\beta_{3}\left(t,x\right)-\beta_{3}\left(t,{\tilde{x}}^{\left(k\right)}\right)}{x_{k}-x_{k}^{\prime }}\right]}^{2} \end{array}\right.$$
and we can easily obtain the relation
(119)$$\left\{\begin{array}{cc} \frac{JD\left(P^{\left(3\right)}\left(t,x\right),P^{\left(3\right)}\left(s,y\right)\right)}{{\left|t-s\right|}^{2}FI_{1}^{\left(t\right)}\left(P^{\left(3\right)}\left(t,x\right)\right)} & =\frac{\beta_{3}\left(t,x\right)}{\beta_{3}\left(s,x\right)}\frac{\beta_{3}\left(t,x\right)+\beta_{3}\left(s,x\right)+4}{2\left[\beta_{3}\left(t,x\right)+2\right]}{\left[\frac{\beta_{3}\left(t,x\right)-\beta_{3}\left(s,y\right)}{b_{0}^{\left(3\right)}\left(t,x\right)\left(t-s\right)}\right]}^{2}\\ \frac{JD\left(P^{\left(3\right)}\left(t,x\right),P^{\left(3\right)}\left(t,{\tilde{x}}^{\left(k\right)}\right)\right)}{|x_{k}-x_{k}^{\prime }{|}^{2}FI_{1}^{\left(x_{k}\right)}\left(P^{\left(3\right)}\left(t,x\right)\right)} & =\frac{\beta_{3}\left(t,x\right)}{\beta_{3}\left(t,{\tilde{x}}^{\left(k\right)}\right)}\frac{\beta_{3}\left(t,x\right)+\beta_{3}\left(t,{\tilde{x}}^{\left(k\right)}\right)+4}{2\left[\beta_{3}\left(t,x\right)+2\right]}{\left[\frac{\beta_{3}\left(t,x\right)-\beta_{3}\left(t,{\tilde{x}}^{\left(k\right)}\right)}{b_{k}^{\left(3\right)}\left(t,x\right)\left(x_{k}-x_{k}^{\prime }\right)}\right]}^{2} \end{array}\right.$$
for k=1,2,⋯,d. Without a loss of generality, the result (Equation 119) also corroborates Theorem 2.

Furthermore, if we consider different $\beta_{3}\left(t,x\right)$ and $\beta_{4}\left(t,x\right)$ in density function (Equation 103), we can obtain density functions $p^{\left(3\right)}\left(u;t,x\right)$ and $p^{\left(4\right)}\left(u;t,x\right)$; then, the generalized Fisher divergence at the same space–time points is
(120)$$\begin{array}{cc} \; & FD_{\left(b_{k}^{\left(3\right)},b_{k}^{\left(4\right)}\right)}\left(P^{\left(3\right)}\left(t,x\right)∥P^{\left(4\right)}\left(t,x\right)\right)\\ = & \frac{\beta_{3}\left(t,x\right)-\beta_{4}\left(t,x\right)}{4\beta_{3}^{2}\left(t,x\right)\beta_{4}^{2}\left(s,y\right)}\\ \; & \times \left\{b_{k}^{\left(4\right)}\left(t,x\right)\beta_{3}\left(t,x\right)\left[\beta_{3}^{2}\left(t,x\right)-2\beta_{4}^{2}\left(s,y\right)\right]-b_{k}^{\left(3\right)}\left(t,x\right)\beta_{4}\left(t,x\right)\left[\beta_{4}^{2}\left(s,y\right)-2\beta_{3}^{2}\left(t,x\right)\right]\right\}\\ \; & \times \left[4\beta_{3}\left(t,x\right)+4\beta_{4}\left(t,x\right)-3\beta_{3}\left(t,x\right)\beta_{4}\left(t,x\right)\right]\left[b_{k}^{\left(4\right)}\left(t,x\right)\beta_{3}\left(t,x\right)-b_{k}^{\left(3\right)}\left(t,x\right)\beta_{4}\left(t,x\right)\right] \end{array}$$
with the remainder terms
(121)$$R_{k}\left(P^{\left(3\right)}\left(t,x\right)∥P^{\left(4\right)}\left(t,x\right)\right)=2\left[b_{k}^{\left(3\right)}\left(t,x\right)-b_{k}^{\left(4\right)}\left(t,x\right)\right]$$k=0,1,2,⋯,d. Then, the generalized De Bruijn identities are as follows:(122)$$\left\{\begin{array}{cc} \frac{\partial }{\partial t}JD\left(P^{\left(3\right)}\left(t,x\right),P^{\left(4\right)}\left(t,x\right)\right)= & -\frac{1}{2}FD_{\left(b_{0}^{\left(3\right)},b_{0}^{\left(4\right)}\right)}\left(P^{\left(3\right)}\left(t,x\right)∥P^{\left(4\right)}\left(t,x\right)\right)\\ \; & -2\left[b_{0}^{\left(3\right)}\left(t,x\right)-b_{0}^{\left(4\right)}\left(t,x\right)\right]\\ \frac{\partial }{\partial x_{k}}JD\left(P^{\left(3\right)}\left(t,x\right),P^{\left(4\right)}\left(t,x\right)\right)= & -\frac{1}{2}FD_{\left(b_{k}^{\left(3\right)},b_{k}^{\left(4\right)}\right)}\left(P^{\left(3\right)}\left(t,x\right)∥P^{\left(4\right)}\left(t,x\right)\right)\\ \; & -2\left[b_{k}^{\left(3\right)}\left(t,x\right)-b_{k}^{\left(4\right)}\left(t,x\right)\right]\\ k= & 1,2,\cdots ,d \end{array}\right.$$

Additionally, we offer an alternative non-trivial form below that utilizes the implicit functions method to express our results. This form differs from the one presented in Theorem 5.

**Theorem** **6.**
*Suppose the functions in Fokker–Planck Equations (Equation 43) for the continuously bounded random field X(t,x)∈[0,1] are formulated as follows:*

(123)
$$\left\{\begin{array}{cc} a_{k}\left(u;t,x\right) & =-\frac{3}{2}b_{k}\left(t,x\right)u\\ b_{k}\left(u;t,x\right) & =b_{k}\left(t,x\right)\left(1-u^{2}\right)\\ k & =0,1,2,\cdots ,d \end{array}\right.$$

*where $b_{k}$ are continuously differentiable functions independent of u and a continuously differentiable function β(t,x) exists such that*

(124)
$$d\beta \left(t,x\right)=b_{0}\left(t,x\right)dt+b_{1}\left(t,x\right)dx_{1}+\cdots +b_{d}\left(t,x\right)dx_{d}$$

*the initial value X(t,x)=0 as prod(t,x)=0 and the initial density function is p(u;t,x)=δ(u) as prod(t,x)=0. Then, the density function is as follows:*

(125)
$$\left\{\begin{array}{cc} p(u;t,x) & =\frac{e^{\frac{1}{2}\beta \left(t,x\right)}}{\sqrt{2\pi \beta (t,x)}}e^{-\frac{v^{2}}{2\beta (t,x)}}\\ u & =\sin v \end{array}\right.$$



**Proof.** Depending on the conditions, it is easy to obtain the Fokker–Planck equations as
(126)$$\left\{\begin{array}{cc} \frac{\partial }{\partial t}p\left(u;t,x\right) & =\frac{b_{0}\left(t,x\right)}{2}\frac{\partial^{2}}{\partial u^{2}}\left[\left(1-u^{2}\right)p\left(u;t,x\right)\right]-\frac{3b_{0}\left(t,x\right)}{2}\frac{\partial }{\partial u}\left[up(u;t,x)\right]\\ \frac{\partial }{\partial x_{k}}p\left(u;t,x\right) & =\frac{b_{k}\left(t,x\right)}{2}\frac{\partial^{2}}{\partial u^{2}}\left[\left(1-u^{2}\right)p\left(u;t,x\right)\right]-\frac{3b_{k}\left(t,x\right)}{2}\frac{\partial }{\partial u}\left[up(u;t,x)\right]\\ k & =1,2,\cdots ,d \end{array}\right.$$By implementing the transformation with u=sinv and defining $\tilde{p}\left(v;t,x\right)=p\left(u\left(v;t,x\right)\right)$, the equations can be restructured as
(127)$$\left\{\begin{array}{cc} \frac{\partial }{\partial t}\tilde{p}\left(v;t,x\right) & =\frac{b_{0}\left(t,x\right)}{2}\frac{\partial^{2}}{\partial v^{2}}\tilde{p}\left(v;t,x\right)+\frac{b_{0}\left(t,x\right)}{2}\tilde{p}\left(v;t,x\right)\\ \frac{\partial }{\partial x_{k}}\tilde{p}\left(v;t,x\right) & =\frac{b_{k}\left(t,x\right)}{2}\frac{\partial^{2}}{\partial v^{2}}\tilde{p}\left(v;t,x\right)+\frac{b_{k}\left(t,x\right)}{2}\tilde{p}\left(v;t,x\right)\\ k & =1,2,\cdots ,d \end{array}\right.$$
with the solution
(128)$$\left\{\begin{array}{cc} \tilde{p}\left(v;t,x\right) & =\frac{e^{\frac{1}{2}\int_{0}^{t}b_{0}\left(s,x\right)ds}}{\sqrt{2\pi \int_{0}^{t}b_{0}\left(s,x\right)ds}}e^{-\frac{v^{2}}{2\int_{0}^{t}b_{0}\left(s,x\right)ds}}\\ \tilde{p}\left(v;t,x\right) & =\frac{e^{\frac{1}{2}\int_{0}^{x_{k}}b_{k}\left(t,x\right)dx_{k}}}{\sqrt{2\pi \int_{0}^{x_{k}}b_{k}\left(t,x\right)dx_{k}}}e^{-\frac{v^{2}}{2\int_{0}^{x_{k}}b_{k}\left(t,x\right)dx_{k}}}\\ k & =1,2,\cdots ,d \end{array}\right.$$Recall that a continuously differential function β(t,x) exists such that
(129)$$d\beta \left(t,x\right)=b_{0}\left(t,x\right)dt+b_{1}\left(t,x\right)dx_{1}+\cdots +b_{d}\left(t,x\right)dx_{d}$$
we can derive the probability density function
(130)$$\tilde{p}\left(v;t,x\right)=\frac{e^{\frac{1}{2}\beta \left(t,x\right)}}{\sqrt{2\pi \beta (t,x)}}e^{-\frac{v^{2}}{2\beta (t,x)}}$$
then,
(131)$$\left\{\begin{array}{cc} p(u;t,x) & =\frac{e^{\frac{1}{2}\beta \left(t,x\right)}}{\sqrt{2\pi \beta (t,x)}}e^{-\frac{v^{2}}{2\beta (t,x)}}\\ u & =\sin v \end{array}\right.$$□

**Remark** **2.**
*Similar to the discussion in Remark 1, we can obtain the Itô processes corresponding to the Fokker–Planck equations in Theorem 6*

(132)
$$\left\{\begin{array}{cc} \partial_{t}X\left(t,x\right) & =-\frac{3}{2}b_{0}\left(t,x\right)X\left(t,x\right)+\sqrt{b_{0}\left(t,x\right)\left[1-X^{2}\left(t,x\right)\right]}W_{t}\\ \partial_{x_{k}}X\left(t,x\right) & =-\frac{3}{2}b_{k}\left(t,x\right)X\left(t,x\right)+\sqrt{b_{k}\left(t,x\right)\left[1-X^{2}\left(t,x\right)\right]}W_{k} \end{array}\right.$$

*k=1,2,⋯,d. In fact, this random field can be solved with a sinusoidal transformation, and the corresponding probability density function can be obtained. Although random field (Equation 132) has not yet found its application scenario, it gives us ideas for constructing different forms on space–time random fields in the future.*


From density function (Equation 125), if we consider different $\beta_{5}\left(t,x\right)$ and $\beta_{6}\left(t,x\right)$, we can obtain density functions $p^{\left(5\right)}\left(u;t,x\right)$ and $p^{\left(6\right)}\left(u;t,x\right)$; then, we can obtain the Jeffreys divergence and generalized Fisher information
(133)$$\begin{array}{c} JD\left(P^{\left(5\right)}\left(t,x\right),P^{\left(5\right)}\left(s,y\right)\right)=\frac{1-\beta_{5}\left(t,x\right)-\beta_{5}\left(s,y\right)}{2\beta_{5}\left(t,x\right)\beta_{5}\left(s,y\right)}{\left[\beta_{5}\left(t,x\right)-\beta_{5}\left(s,y\right)\right]}^{2} \end{array}$$
and
(134)$$\left\{\begin{array}{cc} FI_{1}^{\left(t\right)}\left(P^{\left(5\right)}\left(t,x\right)\right) & =\frac{1-2\beta_{5}\left(t,x\right)}{2\beta_{5}^{2}\left(t,x\right)}{\left[b_{0}^{\left(5\right)}\left(t,x\right)\right]}^{2}\\ FI_{1}^{\left(x_{k}\right)}\left(P^{\left(5\right)}\left(t,x\right)\right) & =\frac{1-2\beta_{5}\left(t,x\right)}{2\beta_{5}^{2}\left(t,x\right)}{\left[b_{k}^{\left(5\right)}\left(t,x\right)\right]}^{2} \end{array}\right.$$
and then the quotients
(135)$$\left\{\begin{array}{cc} \frac{JD\left(P^{\left(5\right)}\left(t,x\right),P^{\left(5\right)}\left(s,x\right)\right)}{{\left|t-s\right|}^{2}} & =\frac{1-\beta_{5}\left(t,x\right)-\beta_{5}\left(s,x\right)}{2\beta_{5}\left(t,x\right)\beta_{5}\left(s,x\right)}{\left[\frac{\beta_{5}\left(t,x\right)-\beta_{5}\left(s,x\right)}{t-s}\right]}^{2}\\ \frac{JD\left(P^{\left(5\right)}\left(t,x\right),P^{\left(5\right)}\left(t,{\tilde{x}}^{\left(k\right)}\right)\right)}{|x_{k}-x_{k}^{\prime }{|}^{2}} & =\frac{1-\beta_{5}\left(t,x\right)-\beta_{5}\left(t,{\tilde{x}}^{\left(k\right)}\right)}{2\beta_{5}\left(t,x\right)\beta_{5}\left(t,{\tilde{x}}^{\left(k\right)}\right)}{\left[\frac{\beta_{5}\left(t,x\right)-\beta_{5}\left(t,{\tilde{x}}^{\left(k\right)}\right)}{x_{k}-x_{k}^{\prime }}\right]}^{2} \end{array}\right.$$

Obviously, we can easily obtain
(136)$$\left\{\begin{array}{cc} \frac{JD\left(P^{\left(5\right)}\left(t,x\right),P^{\left(5\right)}\left(s,x\right)\right)}{{\left|t-s\right|}^{2}FI_{1}^{\left(t\right)}\left(P^{\left(5\right)}\left(t,x\right)\right)} & =\frac{1-\beta_{5}\left(t,x\right)-\beta_{5}\left(s,x\right)}{1-2\beta_{5}\left(t,x\right)}\frac{\beta_{5}\left(t,x\right)}{\beta_{5}\left(s,x\right)}{\left[\frac{\beta_{5}\left(t,x\right)-\beta_{5}\left(s,x\right)}{b_{0}^{\left(5\right)}\left(t,x\right)\left(t-s\right)}\right]}^{2}\\ \frac{JD\left(P^{\left(5\right)}\left(t,x\right),P^{\left(5\right)}\left(t,{\tilde{x}}^{\left(k\right)}\right)\right)}{|x_{k}-x_{k}^{\prime }{|}^{2}FI_{1}^{\left(x_{k}\right)}\left(P^{\left(5\right)}\left(t,x\right)\right)} & =\frac{1-\beta_{5}\left(t,x\right)-\beta_{5}\left(t,{\tilde{x}}^{\left(k\right)}\right)}{1-2\beta_{5}\left(t,x\right)}\frac{\beta_{5}\left(t,x\right)}{\beta_{5}\left(t,{\tilde{x}}^{\left(k\right)}\right)}{\left[\frac{\beta_{5}\left(t,x\right)-\beta_{5}\left(t,{\tilde{x}}^{\left(k\right)}\right)}{b_{k}^{\left(5\right)}\left(t,x\right)\left(x-x_{k}^{\prime }\right)}\right]}^{2} \end{array}\right.$$
for k=1,2,⋯,d. Without a loss of generality, the result (Equation 136) corroborates Theorem 2.

Furthermore, if we consider different $\beta_{5}\left(t,x\right)$ and $\beta_{6}\left(t,x\right)$ in the density function (Equation 125), denoted as $p^{\left(5\right)}\left(u;t,x\right)$ and $p^{\left(6\right)}\left(u;t,x\right)$, we can obtain the generalized Fisher divergence at the same space–time points
(137)$$\begin{array}{cc} \; & FD_{\left(b_{k}^{\left(5\right)},b_{k}^{\left(6\right)}\right)}\left(P^{\left(5\right)}\left(t,x\right)∥P^{\left(6\right)}\left(t,x\right)\right)\\ = & \frac{\left[\beta_{5}\left(t,x\right)-\beta_{6}\left(t,x\right)\right]\left[b_{k}^{\left(6\right)}\beta_{5}\left(t,x\right)-b_{k}^{\left(5\right)}\beta_{6}\left(t,x\right)\right]}{2\beta_{5}^{2}\left(t,x\right)\beta_{6}^{2}\left(t,x\right)}\left[\beta_{5}\left(t,x\right)-\beta_{5}^{2}\left(t,x\right)+\beta_{6}\left(t,x\right)-\beta_{6}^{2}\left(t,x\right)\right] \end{array}$$
with the remainder terms
(138)$$R_{k}\left(P^{\left(5\right)}\left(t,x\right)∥P^{\left(6\right)}\left(t,x\right)\right)=b_{k}^{\left(5\right)}\left(t,x\right)-b_{k}^{\left(6\right)}\left(t,x\right)$$
for k=0,1,2,⋯,d. Then, the generalized De Bruijn identities are as follows:(139)$$\left\{\begin{array}{cc} \frac{\partial }{\partial t}JD\left(P^{\left(5\right)}\left(t,x\right),P^{\left(6\right)}\left(t,x\right)\right) & =-\frac{1}{2}FD_{\left(b_{0}^{\left(5\right)},b_{0}^{\left(6\right)}\right)}\left(P^{\left(5\right)}\left(t,x\right)∥P^{\left(6\right)}\left(t,x\right)\right)-\left[b_{0}^{\left(5\right)}\left(t,x\right)-b_{0}^{\left(6\right)}\left(t,x\right)\right]\\ \frac{\partial }{\partial x_{k}}JD\left(P^{\left(5\right)}\left(t,x\right),P^{\left(6\right)}\left(t,x\right)\right) & =-\frac{1}{2}FD_{\left(b_{k}^{\left(5\right)},b_{k}^{\left(6\right)}\right)}\left(P^{\left(5\right)}\left(t,x\right)∥P^{\left(6\right)}\left(t,x\right)\right)-\left[b_{k}^{\left(5\right)}\left(t,x\right)-b_{k}^{\left(6\right)}\left(t,x\right)\right]\\ k & =1,2,\cdots ,d \end{array}\right.$$

## 5. Conclusions

In this paper, we provide a generalization of the classical definitions of entropy, divergence, and Fisher information and derive these measures on a space–time random field. In addition, the Fokker–Planck Equations (Equation 43) for the space–time random field and density functions are obtained. Moreover, we obtain the Jeffreys divergence of a space–time random field at different space–time positions, and we obtain the approximation of the ratio of Jeffreys divergence to the square of space–time coordinate difference to the generalized Fisher information (Equation 54). Additionally, we use the Jeffreys divergence on two space–time random fields from the same type but different parameters Fokker–Planck equations, to obtain generalized De Bruijn identities (Equation 61). Finally, we give three examples of Fokker–Planck equations, with their solutions, to calculate the corresponding Jeffreys divergence, generalized Fisher information, and Fisher divergence and obtain the De Bruijn identities. These results encourage further research into the entropy divergence of space–time random fields, which advances the pertinent fields of information entropy, Fisher information, and De Bruijn identities.

## Data Availability

Not applicable.

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
