# Peer review of "Jeffreys Divergence and Generalized Fisher Information Measures on Fokker–Planck Space–Time Random Field"

_entropy, 2023, doi:10.3390/e25101445_

Round 1
Reviewer 1 Report
The work aim focuses on Jeffreys Divergence and Generalized Fisher Information. The paper appears well structured and meaningsfull in its content. To my view-point the manuscript could be published in the current form except to some clarifications that could help to improve the soudness of the contribution to more large audience.
More specifically, it may be appropriate to place a strong emphasis on the practical ramifications of the examined method in many and different interest-fields, such as the evaluation and analysis raim. Whit regard multicriteria analysis is there any suitable linkages? Some insights from this kink might help support a global picture of the work.
Reviewer 2 Report
The Author employs Fokker-Planck equations for a space-time random field to investigate associated Jeffreys divergence and generalized Fisher information. The Author proposes multiple theorems and present several examples. Generally, the topic studied by the Author is interesting. But the paper is lacking rigor to view many of the results proposed as rigorous theorems. In particular, the index sets for the process studied in the paper change throughout the paper so that the examples are not necessarily supported by the theorems. Besides, all examples, except for the one presented in Section 4.1, appear to be artificial and not supported by applications. In any case, no appropriate comments are made.
In addition to extensive language, the following issues need to be addressed:
l 32: "less overlap between the two distributions, the result is infinity" - vague language
l. 59-60: "is often used to find the density function" - confusing language
l 70: "derivative with respect to $t$"
throughout the manuscript: "derivable" -> "differentiable"
l 95: The notation is incorrect. Partially differentiable functions cannot be decomposed into a Cartesian product.
l 98: "we denote that" - confusing language
l 128: "arbitrary non-negative function" - needs to be measurable at least to interpret the integral
l 128: Proposition 4 requires lots of assumptions to perform various operations (integration, differentiation, etc.) in the proof.
l 135: "are never vanish" should read "never vanish"
l 146 and ff: "Generalized cross Fisher information" - why capitalized "Generalized"?
l 148 and ff: "is formed as" -> "is defined as"
l 149 and ff: "it’s" -> "it is", etc.
ll 164-165: "can be rewrite as" -> "can be rewritten as"
l 189: "continuous derivable" -> "continuously differentiable". Also, the assumptions of Lemma 1 are obviously incomplete as respective (sub)integrals (Lebesgue or Riemann?) may be infinite or even fail to exist. Appropriate regularity, e.g., expressed in terms of Sobolev regularity, is required.
l 196: The lemma is not rigorous. Appropriate assumptions are required.
l 199: Fourier transform requires appropriate regularity as it does not exist for arbitrary functions.
l 207: Additional conditions are needed.
l 211: "orthonormal basis of the standard identity" - bad English
l 211: "e_k"'s don't seem to appear in the theorem. The proof requires lots of assumptions not stated in the theorem.
l 225: The formulation of Theorem 4 is impossible to follow. The proof is not rigorous. What is "infinitesimal"? The author does not seem to be using non-standard analysis in the paper. How do $p(u; t, x)$, etc., depend on $X(t, x)$?
l 242: Similar issues for Theorem 5.
l 268: "If we let" -> "If we assume"
l 282 and ff: In all theorems, $x, y$ were assume to be in $\mathbb{R}$. Later the Author switches to $\mathbb{R}_{+}$ or $[0, 1]^{n}$ without arguing why the theorems still hold. Note that the latter sets have boundary.
l 323: In Section 4.2, no assumptions on the functions involved are stated.
l 350: "An Interesting Equation" -> that's up to the reader to decide...
Overall, the referee suggests a major revision.
Substantial language editing is required.
Reviewer 3 Report
I think the paper nicely generalizes known results.
I suggest to read the paper once again by the Author since for instance:
0) language need some improvement for instancepage 6 line 164 "can be rewrite as...." I think
it should be "rewritten"
1) What is wrong with Lemma 2 on page 9? It is lemma or part of the proof.
Next
2) page 9 I don't understand Equation (21) and I think I might need some explanation, the same
applies to Equation 35.
3) Lemma 1 I don't understand the statement "holds true if it makes sense"
After carefull reading the paper by the Author, corrections adn theses explanation
I suggest to accept the paper. However at the moment I suggest major revision of the paper.
English can be improves
Round 2
Reviewer 2 Report
The Author was able to adequately address the concerns expressed by the reviewer in his original report.
The manuscript could use some language editing.
Reviewer 3 Report
I suggest to accept the paper after corrections.